# Activation of the EGFR/MAPK pathway drives transdifferentiation of quiescent niche cells to stem cells in the *Drosophila* testis niche

Leah J Greenspan[1,2], Margaret de Cuevas[1]*, Kathy H Le[1,3], Jennifer M Viveiros[1], Erika L Matunis[1]*

[1]Department of Cell Biology, Johns Hopkins University School of Medicine, Baltimore, United States; [2]Division of Developmental Biology, Eunice Kennedy Shriver National Institute of Child Health and Human Development, National Institutes of Health, Bethesda, United States; [3]Graduate Program in Biophysics, Stanford University, Stanford, United States

**Abstract** Adult stem cells are maintained in niches, specialized microenvironments that regulate their self-renewal and differentiation. In the adult *Drosophila* testis stem cell niche, somatic hub cells produce signals that regulate adjacent germline stem cells (GSCs) and somatic cyst stem cells (CySCs). Hub cells are normally quiescent, but after complete genetic ablation of CySCs, they can proliferate and transdifferentiate into new CySCs. Here we find that Epidermal growth factor receptor (EGFR) signaling is upregulated in hub cells after CySC ablation and that the ability of testes to recover from ablation is inhibited by reduced EGFR signaling. In addition, activation of the EGFR pathway in hub cells is sufficient to induce their proliferation and transdifferentiation into CySCs. We propose that EGFR signaling, which is normally required in adult cyst cells, is actively inhibited in adult hub cells to maintain their fate but is repurposed to drive stem cell regeneration after CySC ablation.

**\*For correspondence:**
decuevas@jhmi.edu (MC);
ematuni1@jhmi.edu (ELM)

**Competing interest:** The authors declare that no competing interests exist.

## Editor's evaluation

In this manuscript, the Authors demonstrate that EGFR signaling plays an important role in somatic cyst stem cells and hub cells in the male germ cell lineage. Using a variety of genetic, biochemical and cell biological approaches, they provide a regulatory frame work for how the hub cells maintain their cell fate.

## Introduction

Stem cells, which are unique in their ability to self-renew and produce daughter cells that differentiate into the mature cells of a tissue, reside in specialized microenvironments, called niches, that contain signals vital for stem cell maintenance (*Drummond-Barbosa, 2019*; *Greenspan et al., 2015*). Loss of a stem cell population due to injury or aging can be detrimental to tissue function, but recent studies have shown that stem cells can be regenerated even after complete loss of a stem cell population (*Beumer and Clevers, 2021*; *Greenspan et al., 2015*). In the *Drosophila* testis, both germline and somatic stem cells can be regenerated de novo by conversion of other cell types into stem cells, which makes this well-characterized system an exceptional model for studying the dynamics of stem cell regeneration in vivo (*Brawley and Matunis, 2004*; *Cheng et al., 2008*; *Greenspan and Matunis,*

*2018*; *Hétié et al., 2014*, *Sheng et al., 2009*; *Voog et al., 2014*). The molecular mechanisms that trigger these events remain largely unknown, but elucidating them could lead to the development of novel regenerative and anti-aging therapies.

The *Drosophila* testis niche contains a cluster of somatic hub cells that sends signals to two types of stem cells: germline stem cells (GSCs), which give rise to sperm, and somatic cyst stem cells (CySCs), which give rise to the cyst cells that encase differentiating germ cells (*Figure 1A*; *Hardy et al., 1979*). CySCs and hub cells derive from a common pool of precursor cells and are specified early in embryo-genesis (*Dinardo et al., 2011*; *Le Bras and Van Doren, 2006*). Hub cells are completely quiescent in adult flies, but after injury to the testis, in the form of genetic ablation of CySCs, they can exit quies-cence, leave the hub, and transdifferentiate into CySCs, thereby replenishing the lost population of stem cells (*Hétié et al., 2014*). The molecular signals that trigger hub cells to re-enter the cell cycle are not known but are likely acting upstream of Cyclin-dependent kinase 4 (Cdk4) and its binding partner Cyclin D, since the forced over-expression of these two proteins is sufficient to drive hub cells out of quiescence and into the transdifferentiation program (*Hétié et al., 2014*). Similarly, conversion of adult hub cells into CySCs also occurs when the Snail-family transcription factor Escargot or the tumor suppressor and Cdk4/Cyclin D target Retinoblastoma-family protein (Rbf) is knocked down directly in hub cells (*Greenspan and Matunis, 2018*; *Voog et al., 2014*). Although many examples of transdifferentiation have been reported in vitro, there are relatively few examples in vivo (*Merrell and Stanger, 2016*), making the upstream events that instruct hub cells to transdifferentiate upon injury to the *Drosophila* testis of interest.

The epidermal growth factor receptor/mitogen-activated protein kinase (EGFR/MAPK) signaling pathway is a conserved receptor tyrosine kinase pathway that regulates many aspects of develop-ment including cell survival, growth, proliferation, and differentiation (*Wee and Wang, 2017*). In the developing *Drosophila* testis, the EGFR signaling pathway is essential for the formation and function of cyst cells. While Notch is expressed in all somatic gonadal precursor cells (SGPs) and required for hub cell fate during embryogenesis, EGFR signaling is only expressed in a subset, repressing hub cell fate and enabling these cells to differentiate into cyst lineage cells (*Kitadate and Kobayashi, 2010*; *Okegbe and DiNardo, 2011*). In adult testes, EGFR signaling is required in cyst lineage cells for their encapsulation of germ cells, and this association is vital for germ cell differentiation and maturation into sperm (*Hudson et al., 2013*; *Kiger et al., 2000*; *Sarkar et al., 2007*; *Schulz et al., 2002*; *Tran et al., 2000*). EGFR signaling is also required in the adult germline stem cells to regulate the frequency of their divisions (*Parrott et al., 2012*). Whether EGFR signaling has a role in adult hub cells has not yet been investigated.

Here, we show that the direct activation of EGFR signaling in adult hub cells is sufficient to cause them to re-enter the cell cycle and transdifferentiate into CySCs. We also provide evidence suggesting that EGFR signaling is required in the hub for testes to recover from CySC ablation. Our results support the hypothesis that EGFR signaling, which normally functions in cyst cells and germ cells in adult testes, is repurposed after complete CySC ablation to drive the de novo formation of stem cells from adjacent quiescent niche cells in an adult tissue.

## Results

### Activation of EGFR signaling in quiescent adult hub cells is sufficient to trigger cell cycle re-entry

To identify changes in signaling that are sufficient to drive hub cells into mitosis, we conducted a genetic screen in which we mis-expressed components of major signaling pathways in the adult hub and immunostained testes for mitotic hub cells (*Figure 1B*). We used the Gal4-UAS system to condi-tionally knock down or over-express candidate genes specifically in adult hub cells; candidates included receptors, downstream effectors, and transcription factors from most of the canonical signaling path-ways in *Drosophila* (*Housden and Perrimon, 2014*). Flies carrying the hub-specific driver *E132-Gal4* and the temperature-sensitive Gal4 inhibitor *tub-Gal80$^{ts}$* (which we abbreviate as *E132-Gal4, Gal80$^{ts}$*) were crossed to flies carrying UAS-based over-expression or knockdown (dominant-negative or RNAi) constructs (*UAS-X*). Crosses were set at permissive temperature (18 °C), and adult male progeny (*E132-Gal4, Gal80$^{ts}$ > UAS X*) were shifted to restrictive temperature (31 °C) for 7 days to induce expression of the transgene. Testes were then dissected, fixed, and immunostained for the hub cell

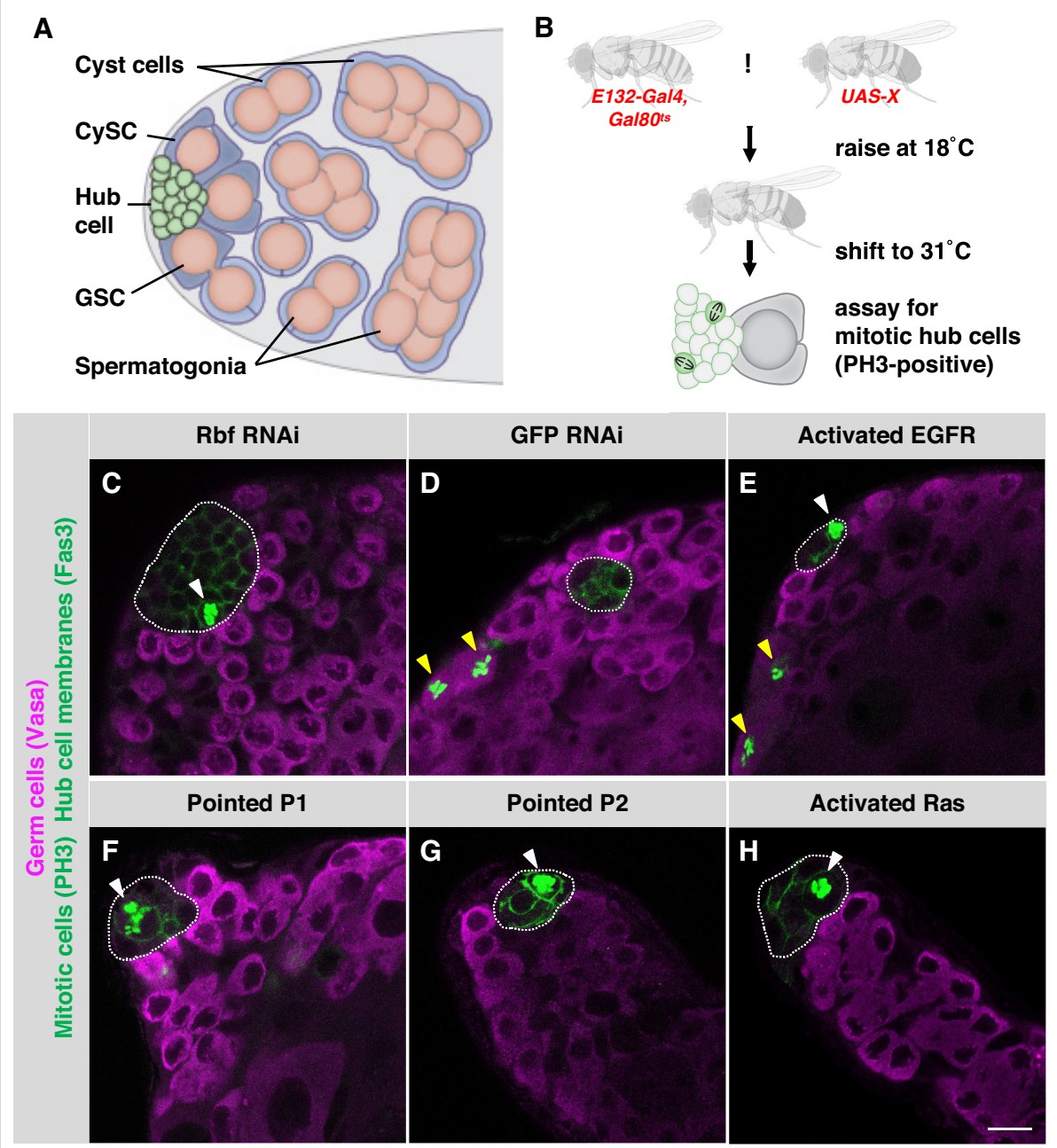

**Figure 1.** Activation of EGFR signaling in adult hub cells causes them to re-enter the cell cycle. (**A**) Schematic of the *Drosophila* testis stem cell niche. Somatic hub cells (green) secrete signals to adjacent germline stem cells (GSCs, orange) and somatic cyst stem cells (CySCs, dark blue). Both types of stem cells divide asymmetrically to produce differentiating daughter cells. Somatic cyst cells (light blue) envelop clusters of spermatogonia (orange), and together they move away from the testis apex as they differentiate. (**B**) Schematic of the screen for signals that can trigger hub cells to re-enter the cell cycle. Candidate signaling pathway genes were conditionally mis-expressed in adult hub cells using the hub specific driver *E132-Gal4*, and mitotic hub cells were identified by immunostaining for phospho-histone H3 (PH3) (see main text for details). (**C–H**) Single confocal sections through the apex of testes immunostained for Fas3 (hub cell membranes, green), PH3 (mitotic chromosomes, nuclear green), and Vasa (germ cells, magenta). Hubs are outlined in white. Mitotic hub cells (white arrowheads) are found in positive control testes (with Rbf knockdown in the hub, **C**) and in testes over-expressing components of the canonical EGFR/MAPK pathway in the hub (**E–H**) but not in negative control testes (**D**). Mitotic cells are also found outside the hub in both control and experimental testes, as expected (yellow arrowheads). Scale bar (in H, for all panels) is 10 μm.

**Table 1.** Hub cell proliferation after EGFR pathway activation.

| Gal4 driver | UAS line (BDSC #) | Days at 31 °C | % Testes with PH3-positive hub cells |
|---|---|---|---|
| **Controls** | | | |
| E132-Gal4, Gal80$^{ts}$ | UAS-Rbf-RNAi (41863) | 7 | **31% (n = 34/108)\*\*\*\*** |
| | UAS-Rbf-RNAi (36744) | 7 | **29% (n = 53/183)\*\*\*\*** |
| | UAS-GFP-RNAi (9330 or 9331) | 7 | 0% (n = 0/237) |
| C587-Gal4, Gal80$^{ts}$ | UAS-GFP-RNAi (9331) | 7 | 0% (n = 0/294) |
| | | | |
| **Downstream Effectors and Transcription Factors** | | | |
| E132-Gal4, Gal80$^{ts}$ | UAS-Pointed.P1 (869) | 7 | **7% (n = 4/55)\*\*** |
| | UAS-Pointed.P2 (399) | 7 | **6% (n = 6/100)\*\*\*** |
| | | 3 | **11% (n = 2/18)\*\*** |
| | UAS-Ras85D.V12 (4847) | 5 | **6% (n = 2/36)\*** |
| | UAS-Rolled (59006) | 7 | 0% (n = 0/92)[ns] |
| **Receptors** | | | |
| E132-Gal4, Gal80$^{ts}$ | UAS-Egfr Type I (9534) | 7 | 1% (n = 1/143)[ns] |
| | UAS-Egfr Type II (9533) | 7 | **4% (n = 5/117)\*\*** |
| | UAS-Egfr $\lambda$ (59843) | 7 | < 1% (n = 1/163)[ns] |
| | UAS-PvR $\lambda$ (58496) | 7 | 1% (n = 1/106)[ns] |
| | UAS-PvR $\lambda$ (58428) | 7 | 0% (n = 0/156)[ns] |
| | UAS-InR (8250) | 7 | 0% (n = 0/48)[ns] |
| | UAS-InR (8263) | 7 | 0% (n = 0/41)[ns] |
| | UAS-Heartless $\lambda$ (5367) | 7 | 2% (n = 3/181)[ns] |
| | UAS-Breathless $\lambda$ (29045) | 7 | 0% (n = 0/90)[ns] |
| **Negative Regulators** | | | |
| E132-Gal4, Gal80$^{ts}$ | UAS-Sprouty-RNAi (36709) | 7 | **3% (n = 4/144)\*** |
| | UAS-Pten-RNAi (33643) | 7 | **3% (n = 4/125)\*** |
| | UAS-Argos-RNAi (28383) | 7–8 | < 1% (n = 1/170)[ns] |
| C587-Gal4, Gal80$^{ts}$ | UAS-Argos-RNAi (28383) | 7–8 | **2% (n = 6/348)\*** |

Percentages in **bold** are significant compared to the negative control (UAS-GFP-RNAi driven by the same Gal4 driver).
Fisher's Exact Test: \*p < 0.05, \*\*p < 0.01, \*\*\*p < 0.001, \*\*\*\*p < 0.0001, ns = not significant.

marker Fasciclin 3 (Fas3) and the mitotic cell marker Phospho-histone H3 (PH3). As a positive control for our candidate screen, we included testes from flies in which hub cells are forced to re-enter the cell cycle by knockdown of the cell cycle inhibitor Rbf (*Greenspan and Matunis, 2018*). As expected, PH3-positive cells were detected in 29–32% of these positive control testes (*E132-Gal4, Gal80$^{ts}$ > UAS-Rbf-RNAi*; n = 291 testes; *Table 1*; *Figure 1C*). Importantly, PH3-positive hub cells were not detected in testes from flies serving as negative controls (*E132-Gal4, Gal80$^{ts}$ > UAS-GFP-RNAi*; n = 237 testes; *Table 1*; *Figure 1D*). PH3-positive hub cells were not detected in testes with knockdown or over-expression of most signaling pathways (*E132-Gal4, Gal80$^{ts}$ > UAS X*; Appendix 1). However, PH3-positive hub cells were detected in significant numbers of testes from several fly lines over-expressing components of the canonical EGFR/MAPK pathway (*Table 1*). Upon ligand binding, EGFR becomes active and positively regulates Ras; activated Ras initiates a signaling cascade that ultimately activates MAPK, which phosphorylates Pointed, the major activating transcription factor of the pathway (reviewed in *Lusk et al., 2017*; *Shilo, 2014*). Over-expression of Pointed (either isoform, P1 or P2; *Klaes et al., 1994*) or a constitutively active form of EGFR in the hub resulted in significant numbers

of testes with PH3-positive hub cells (7% of testes, n = 55, for P1; 6% of testes, n = 100, for P2; or 4% of testes, n = 117, for EGFR; *Figure 1E–G*). Over-expression of constitutively active Ras for 7 days caused testes to develop grossly abnormal or missing hubs (data not shown), complicating the scoring process; however, we found significant numbers of testes with PH3-positive hub cells after a shorter period (3 or 5 days) of constitutively active Ras expression in the hub (11% of testes, n = 18, after 3 days, or 6% of testes, n = 36, after 5 days; *Figure 1H*). The percentage of testes with PH3-positive hub cells upon over-expression of EGFR/MAPK pathway components (*Table 1*) is similar to that seen during recovery from CySC ablation (6–10% testes have PH3-positive hub cells at 1–5 days after ablation; *Hétié et al., 2014*). We conclude that activation of EGFR signaling in the hub is sufficient to drive quiescent hub cells into mitosis.

In *Drosophila*, the MAPK pathway can potentially be activated not only by EGFR but also by other receptor tyrosine kinases (*Shilo, 2014*). We therefore asked if over-expressing constitutively active forms of other receptor tyrosine kinases in adult hub cells can also cause them to re-enter the cell cycle. We again used *E132-Gal4, Gal80^ts* to drive hub-specific expression of activated forms of the two *Drosophila* Fibroblast growth factor receptors (Heartless and Breathless), the PDGF- and VEGF-receptor related (Pvr); and the Insulin-like receptor (InR). Although a few testes acquired PH3-positive hub cells after over-expression of activated Heartless or Pvr, these results were not statistically significant compared to the negative control (*Table 1*), and we found no PH3-positive hub cells after over-expressing activated Breathless or InR. These results suggest that activation of EGFR within hub cells plays a unique role in forcing these cells to lose quiescence and re-enter the cell cycle.

## EGFR is not required in adult hub cells for their maintenance

Having found that activation of EGFR in adult hub cells is sufficient to drive them into mitosis, we next asked whether EGFR is normally required in these cells, as it is in other cell types in the adult testis (*Hudson et al., 2013*; *Kiger et al., 2000*; *Parrott et al., 2012*; *Sarkar et al., 2007*; *Schulz et al., 2002*; *Tran et al., 2000*). We again used *E132-Gal4, Gal80^ts* to conditionally express EGFR dominant negative or RNAi constructs in adult hub cells. Crosses were set at 18 °C and adult male progeny were shifted to 31 °C for 7 days to induce expression of the *UAS-X* construct. We tested one EGFR dominant-negative line and four RNAi lines, and in all cases these experimental testes were indistinguishable from our negative control testes (*E132-Gal4, Gal80^ts > UAS-GFP-RNAi*; *Table 2*). We verified that each knockdown construct was working as expected by crossing each line to the cyst lineage driver *C587-Gal4* together with *tub-Gal80^ts* (which we abbreviate as *C587-Gal4, Gal80^ts*) and driving expression for 1–2 weeks in adult cyst cells. With all five lines, testes showed a strong phenotype with accumulation of early germ cells in many or all testes (data not shown), as shown previously by *Hudson et al., 2013*. We therefore conclude that adult hub cells in otherwise unperturbed testes do not cell-autonomously require EGFR.

## Notch is not required to maintain adult hub cell quiescence

In the male embryonic gonad, EGFR and Notch act antagonistically to regulate development of the hub. Hub and cyst lineage cells both differentiate from a pool of somatic gonadal precursor cells (SGPs). Notch is activated in almost all SGPs and is required to specify hub cell identity, whereas in posterior SGPs, hub cell fate is repressed by activation of EGFR, causing these cells to differentiate into cyst lineage cells instead (*Kitadate and Kobayashi, 2010*; *Okegbe and DiNardo, 2011*). To determine if Notch plays a role in the adult hub, we conditionally expressed constitutively active (CA), dominant negative (DN), or RNAi knockdown constructs of Notch in adult hub cells using *E132-Gal4, Gal80^ts*. No PH3-positive hub cells or any other obvious hub phenotypes were detected with any line after 7 days of transgene expression (n = 48 testes for CA, 54 testes for DN, or 26–119 testes for RNAi; *Table 2*). We then asked if altered levels of Notch could affect the number of dividing hub cells upon activation of EGFR in the adult hub. We conditionally expressed activated EGFR together with a constitutively active or dominant negative form of Notch in hub cells using *E132-Gal4, Gal80^ts*, but we found no significant difference in the number of PH3-positive hub cells in either case compared to activated EGFR by itself (*Table 2*). These results suggest that Notch does not play a role in maintaining hub cell quiescence in the adult testis.

**Table 2.** Hub cell proliferation upon EGFR and Notch signaling changes.

| UAS lines (with E132-Gal4, Gal80$^{ts}$) | Source | Days at 31 °C | % Testes with PH3-positive hub cells |
|---|---|---|---|
| **Control** | | | |
| UAS-GFP-RNAi | BDSC 9330 or 9331 | 7 | 0% (n = 0/237)* |
| **Egfr knockdown** | | | |
| UAS-Egfr-DN | BDSC 5364 | | 0% (n = 0/103)[†,ns] |
| UAS-Egfr-RNAi | BDSC 36770 | | 0% (n = 0/45)[†,ns] |
| UAS-Egfr-RNAi | BDSC 60012 | 7 | 0% (n = 0/46)[†,ns] |
| UAS-Egfr-RNAi | VDRC 43267 | | 0% (n = 0/29)[†,ns] |
| UAS-Egfr-RNAi[††] | VDRC 107130 | | 0% (n = 0/43)[†,ns] |
| **Notch over-expression or knockdown** | | | |
| UAS-N-CA | BDSC 52008 | | 0% (n = 0/48)[†,ns] |
| UAS-N-DN | BDSC 51667 | | 0% (n = 0/54)[†,ns] |
| UAS-N-RNAi | BDSC 33611 | | 0% (n = 0/119)[†,ns] |
| UAS-N-RNAi | BDSC 33616 | 7 | 0% (n = 0/63)[†,ns] |
| UAS-N-RNAi | BDSC 35213 | | 0% (n = 0/29)[†,ns] |
| UAS-N-RNAi | BDSC 35640 | | 0% (n = 0/26)[†,ns] |
| **Egfr over-expression and Notch combinations** | | | |
| UAS-Egfr Type II | BDSC 9533 | 7 | 4% (n = 5/117)* |
| UAS-EGFR Type I; UAS-EGFR Type II | BDSC 9533 + 9534 | | 4% (n = 11/301)[‡,ns] |
| UAS-N-DN; UAS-EGFR Type II | BDSC 9533 + 51667 | 7–8 | 3% (n = 4/138)[‡,ns; §,ns] |
| UAS-N-CA; UAS-EGFR Type II | BDSC 9533 + 52008 | | 6% (n = 12/205)[‡,ns; §,ns] |
| **Sprouty knockdown and Notch combinations** | | | |
| UAS-Sprouty-RNAi | BDSC 36709 | 7 | 3% (n = 4/144)* |
| UAS-EGFR Type I; UAS-Sprouty-RNAi | BDSC 36709 + 9534 | | 3% (n = 3/101)[¶,ns] |
| UAS-N-DN; UAS-Sprouty-RNAi | BDSC 36709 + 51667 | 7–8 | 1% (n = 1/129)[¶,ns; **,ns] |
| UAS-N-CA; UAS-Sprouty-RNAi | BDSC 36709 + 52008 | | 0% (n = 0/80) [¶,ns; **,ns] |

Fisher's Exact Test: ns = not significant.

*Data from **Table 1**. [††]Another UAS-Egfr-RNAi line, BDSC 36773, did not show a phenotype with the control driver (C587-Gal4) and is not included here.

[†]Compared with UAS-GFP RNAi.

[‡]Compared with UAS-EGFR Type II.

[§]Compared with UAS-EGFR Type I; UAS-EGFR Type II.

[¶]Compared with UAS-Sprouty RNAi.

[**]Compared with UAS-EGFR Type I; UAS-Sprouty RNAi.

## Activation of EGFR signaling in adult hub cells causes them to transdifferentiate into CySCs

In adult flies, genetic ablation of all CySCs, or knockdown of Rbf in the hub, causes hub cells not only to re-enter the cell cycle but also to leave the hub and transdifferentiate into CySCs (*Greenspan and Matunis, 2018*; *Hétié et al., 2014*). To test if EGFR activation in hub cells is sufficient to drive both hub cell proliferation and transdifferentiation, we performed lineage analysis of hub cells using the Gal4 Technique for Real-time and Clonal Expression (G-TRACE; *Evans et al., 2009*). This system, which we used previously to mark cyst lineage cells that arise from transdifferentiating hub cells upon knockdown of Rbf (*Greenspan and Matunis, 2018*), marks the nuclei of cells currently expressing Gal4 with a red fluorescent protein (RFP) and permanently marks the nuclei of any cells originating from the Gal4-expressing cells with a green fluorescent protein (GFP). We drove expression of EGFR pathway components (*UAS-X*) and G-TRACE in adult hub cells using *E132-Gal4, Gal80$^{ts}$*. Flies were

**Table 3.** Hub cell fate conversion after EGFR pathway activation.

| Gal4 driver | UAS lines (BDSC #) | Days at 29 °C | % Testes with GFP-marked cells outside the hub |
|---|---|---|---|
| E132-Gal4, Gal80$^{ts}$ | UAS-G-TRACE (28280) | 8 | 0% (n = 0/39) |
|  | UAS-G-TRACE (28280); UAS-Egfr Type II (9533) |  | **61% (n = 19/31)****** |
|  | UAS-G-TRACE (28280); UAS-Pointed.P1 (869) |  | **38% (n = 19/50)****** |
|  | UAS-G-TRACE (28280); UAS-Pointed.P2 (399) |  | **22% (n = 4/18)**** |
|  | UAS-G-TRACE (28280); UAS-Sprouty-RNAi (36709) |  | **36% (n = 32/89)****** |
|  | UAS-G-TRACE (28280); UAS- Pten-RNAi (33643) |  | **18% (n = 6/34)**** |
| E132-Gal4, Gal80$^{ts}$ | UAS-G-TRACE (28281) | 8 | 0% (n = 0/84) |
|  | UAS-Egfr Type I (9534); UAS-G-TRACE (28281) |  | 2% (n = 1/47)$^{ns}$ |

Percentages in **bold** are significant compared to the negative control (corresponding UAS-G-TRACE alone).
Fisher's Exact Test: **p < 0.01, ****p < 0.0001, ns = not significant.

raised at 18 °C and then shifted to 29 °C for 8 days to induce simultaneous expression of *UAS-X* and *G-TRACE*. (At 31 °C, we found high background marking with the G-TRACE system alone, necessitating a lower temperature for this experiment.) In control testes expressing the G-TRACE cassette alone (*E132-Gal4, Gal80$^{ts}$ > G-TRACE*), hub cell nuclei were marked with both RFP (real-time Gal4 expression) and GFP (permanent marking expression), but cyst lineage cells outside the hub remained unmarked, as expected (*Table 3*; *Figure 2A*). In testes over-expressing components of the EGFR/MAPK pathway in the hub together with G-TRACE, hub cell nuclei were again marked with RFP and GFP, but we also found GFP-marked nuclei outside the hub in significant numbers of testes (*Table 3*; *Figure 2B–D*). Most of the GFP-marked nuclei outside the hub had little to no detectable RFP, indicating that the *E132-Gal4* hub cell driver was no longer expressed in these cells. This observation, together with the morphology and arrangement of these nuclei, suggests that the GFP-marked cells outside the hub are cyst lineage cells that have lost their hub cell identity. We conclude that activation of EGFR signaling in adult hub cells is sufficient to cause their conversion to cyst lineage cells.

## Negative regulation of EGFR maintains hub cell quiescence and identity in adult testes

Several negative regulators of the EGFR pathway have been identified in *Drosophila* (reviewed in *Shilo, 2014*). One of these is Sprouty, a cytoplasmic protein that is expressed in response to EGFR signaling and inhibits the pathway by a negative-feedback mechanism inside the signal-receiving cell (*Casci et al., 1999*; *Hacohen et al., 1998*; *Kramer et al., 1999*). Since activation of EGFR signaling in hub cells is sufficient to trigger their transdifferentiation, we hypothesized that Sprouty might be cell-autonomously required to maintain quiescence in adult hub cells. To test this hypothesis, we used *E132-Gal4, Gal80$^{ts}$* to drive expression of a Sprouty RNAi knockdown construct (*UAS-sprouty-RNAi*) in adult hub cells. After 7 days of knockdown, a significant number of testes had PH3-positive hub cells (3% of testes, n = 144; *Table 1*; *Figure 3A*), confirming that Sprouty is required in adult hub cells to maintain their quiescence. To determine if hub cells depleted for Sprouty can also transdifferentiate into CySCs, we expressed Sprouty RNAi together with G-TRACE in adult hub cells (*E132-Gal4, Gal80$^{ts}$ > UAS-sprouty-RNAi + G-TRACE*). After 8 days, a significant number of testes had GFP-marked cells outside the hub (*Table 3*; *Figure 3E*), indicating that knockdown of Sprouty in adult hub cells is sufficient to cause their conversion to cyst lineage cells.

In addition to Sprouty, Phosphatase and tensin homolog (PTEN), best known as an inhibitor of the phosphoinositide 3-kinase (PI3K) pathway, has been shown to inhibit EGFR within signal-receiving cells, in this case by promoting EGFR degradation (*Vivanco et al., 2010*). Therefore, we asked whether PTEN is required in adult hub cells. After 7 days of PTEN knockdown in the hub (*E132-Gal4, Gal80$^{ts}$ > UAS-Pten-RNAi*), a significant number of testes had PH3-positive hub cells (3% of testes, n = 125; *Table 1*; *Figure 3B*). Knockdown of PTEN in adult hub cells is also sufficient to cause their conversion to cyst lineage cells: after expressing PTEN RNAi together with G-TRACE in adult hub cells for 7 days (*E132-Gal4, Gal80$^{ts}$ > UAS-Pten-RNAi + G-TRACE*), we found a significant

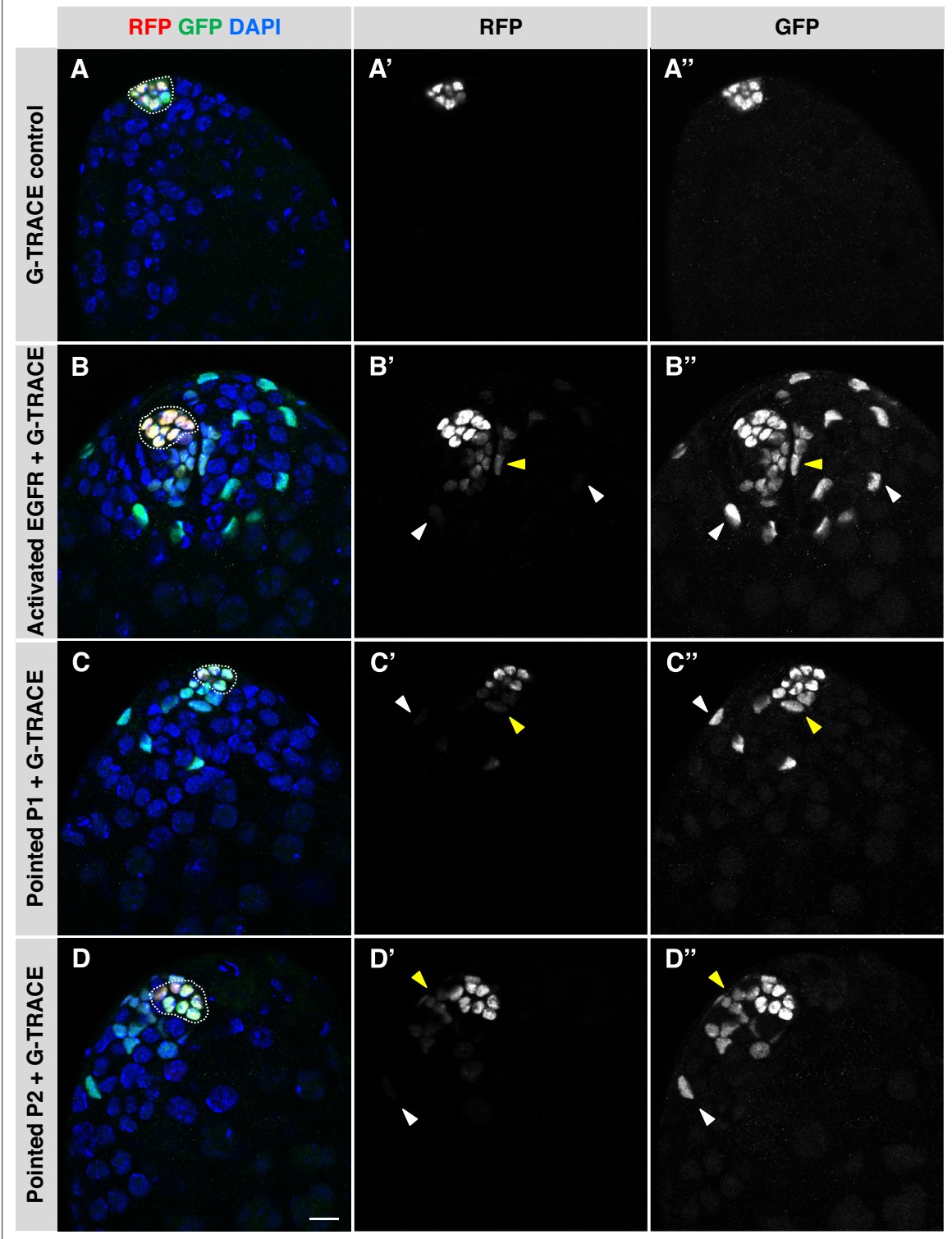

**Figure 2.** Activation of EGFR signaling in adult hub cells causes them to convert to CySCs. (**A–D**) Single confocal sections through the apex of testes after 8 days of G-TRACE lineage tracing system expression in adult hub cells. Testes were immunostained for RFP (red fluorescent protein, marking current expression of the hub-specific driver *E132-Gal4*) and GFP (green fluorescent protein, marking both current and past expression of *E132-Gal4*) and counterstained with DAPI (blue; marks all nuclei). Hubs are outlined in white. (**A'-D'**) Red channel alone, in white; (**A"-D"**) green channel alone, in white. In control testes (**A**), hub cells expressing the G-TRACE system alone are marked with RFP and GFP, but no cells outside the hub are marked. In testes over-expressing components of the EGFR signaling pathway in the hub together with G-TRACE (**B–D**), hub cells are marked with RFP and GFP, and cells outside the hub are also marked, either with GFP and low levels of RFP (yellow arrowheads) or with GFP only (white arrowheads). The marked

*Figure 2 continued on next page*

*Figure 2 continued*

cells outside the hub appear to be cyst lineage cells that arose by conversion of hub cells to CySCs, which no longer express RFP once they lose their hub cell fate. Scale bar in D, for all panels, is 10 μm.

number of testes with GFP-marked cells outside the hub (*Table 3*; *Figure 3F*). We conclude that PTEN is required in adult hub cells to maintain their quiescence and prevent their conversion to cyst lineage cells. We did not find PH3-positive hub cells after over-expressing or knocking down other members of the canonical PI3K pathway (*Table 1*; Appendix 1), which supports the idea that PTEN acts in hub cells by inhibiting EGFR rather than by regulating the PI3K pathway. As further support for this hypothesis, we assayed for EGFR pathway activation using di-phosphorylated ERK (dpERK) as an indicator. Two regulatory residues in ERK/MAPK (encoded by the *rolled* gene in *Drosophila*) are phosphorylated upon pathway activation and can be detected by antibodies specific for dpERK (*Biggs et al., 1994*; *Gabay et al., 1997*). We assessed dpERK immunostaining levels in hub cells in flies conditionally expressing PTEN RNAi in the hub (*E132-Gal4, Gal80ts > UAS-Pten-RNAi*) or conditionally expressing Sprouty RNAi in the hub (*E132-Gal4, Gal80ts > UAS-sprouty-RNAi*) and compared them to control flies with no EGFR pathway perturbations (*y w*). After 7 days at restrictive temperature (31 °C), PTEN and Sprouty knockdown flies both had significantly higher levels of dpERK in the hub than control (*y w*) flies (*Figure 3C*), confirming that the EGFR/MAPK pathway is up-regulated in hub cells upon knockdown of PTEN or Sprouty. Taken together, these results support the hypothesis that the EGFR pathway is actively inhibited in adult hub cells by at least two cell-autonomous inhibitors, Sprouty and PTEN, which maintain hub cell quiescence and prevent their conversion to cyst lineage cells.

The final negative regulator of the EGFR pathway we examined, Argos, is a secreted protein that acts extracellularly by binding to and sequestering the EGF ligand Spitz (*Golembo et al., 1996*; *Klein et al., 2004*; *Schweitzer et al., 1995*). To ask if Argos is required cell-autonomously to maintain hub cell quiescence, we drove expression of an Argos RNAi knockdown construct in adult hub cells for 7 days (*E132-Gal4, Gal80ts > UAS-argos-RNAi*), but we did not find a significant number of testes with PH3-positive hub cells in this case ( < 1% of testes, n = 170; *Table 1*). However, Argos is induced by EGFR signaling (*Golembo et al., 1996*), which occurs at high levels in cyst lineage cells in adult testes (*Chen et al., 2013*; *Fairchild et al., 2016*; *Schulz et al., 2002*), suggesting that cyst lineage cells could be a source of Argos in the testis. To test this hypothesis, we drove expression of *UAS-argos-RNAi* conditionally in adult testes using the cyst lineage driver *C587-Gal4, Gal80ts* (*C587-Gal4, Gal80ts > UAS-argos-RNAi*). After 7 days of knockdown, we found a significant number of testes with PH3-positive hub cells (2% of testes, n = 287; *Table 1*; *Figure 3D*), supporting the hypothesis that Argos is required non-cell-autonomously to maintain hub cell quiescence in adult testes. For technical reasons, we are unable to drive expression of G-TRACE in hub cells while knocking down Argos in cyst lineage cells, so we cannot assess hub cell transdifferentiation in this case.

Having found that hub cells can be forced to re-enter the cell cycle by EGFR pathway activation - via over-expression of activators or knockdown of inhibitors - we next asked if over-expression of EGF ligands is sufficient to induce the same phenotype. Four EGF ligands have been identified in *Drosophila*: Gurken, Keren, Spitz, and Vein. Vein is produced as a secreted protein, but the other three are produced as inactive membrane precursors that must be cleaved to produce the active, secreted form of the ligand (reviewed in *Shilo, 2014*). We over-expressed Vein or secreted forms of the other ligands using *E132-Gal4, Gal80ts* or *C587-Gal4, Gal80ts* to conditionally drive expression in adult hub cells or cyst lineage cells, respectively. After 6–9 days of ligand over-expression, we did not find a significant number of testes with PH3-positive hub cells with any of these fly strains (*Table 4*). We speculated that the levels of EGF produced by these transgenes might not be high enough to override negative regulation by pathway inhibitors, and to test this hypothesis, we drove expression of *UAS-secreted Spitz* and *UAS-argos-RNAi* together in adult testes using *C587-Gal4, Gal80ts* (*C587-Gal4, Gal80ts > UAS-argos-RNAi + UAS-secreted Spitz*). However, the number of testes with PH3-positive hub cells in these flies was not significantly different than in flies expressing *UAS-argos-RNAi* alone (*Table 4*). We conclude that over-expression of EGF ligands is not sufficient to trigger loss of hub cell quiescence, or to enhance the loss of quiescence caused by knockdown of the Spitz antagonist Argos, but the reason for these results is not clear.

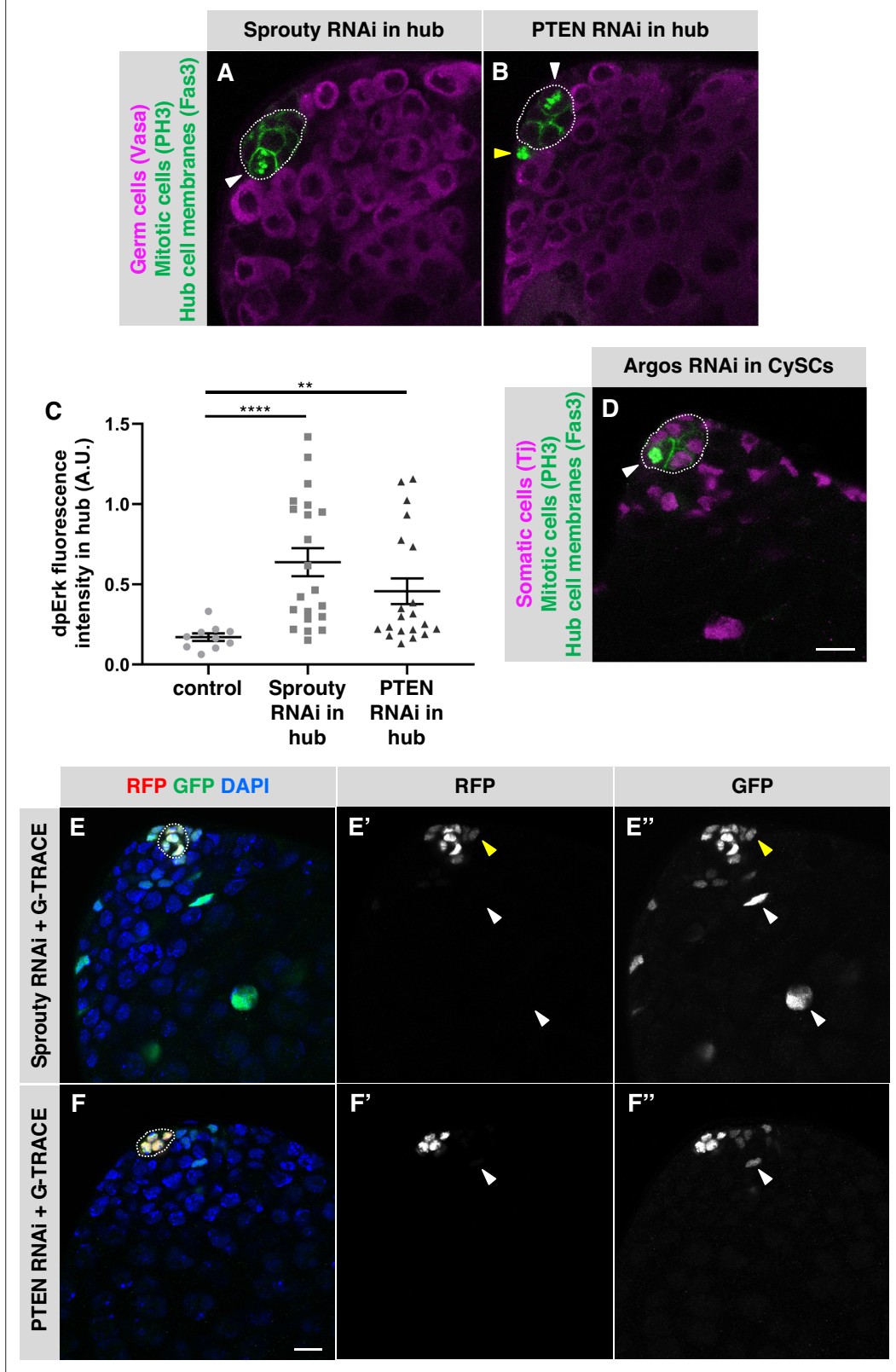

**Figure 3.** Negative regulators of EGFR signaling maintain hub cell quiescence and identity. (**A–B, D**) Single confocal sections through the apex of testes immunostained for Fas3 (hub cell membranes, green), PH3 (mitotic chromosomes, nuclear green), and either (**A–B**) Vasa (germ cells, magenta) or (**D**) Traffic jam (Tj; somatic cell nuclei, magenta). Hubs are outlined in white. Mitotic hub cells (white arrowheads) are found after knockdown of Sprouty

*Figure 3 continued on next page*

*Figure 3 continued*

(**A**) or PTEN (**B**) in the hub or after knockdown of Argos (**D**) in cyst lineage cells. Mitotic cells outside the hub are also found (yellow arrowhead). (**C**) Quantification of dpERK levels in the hub in control, Sprouty knockdown, or Pten knockdown in the hub. dpErk levels in the hub are significantly higher when either Sprouty or Pten are knocked down in the hub than in control testes suggesting these proteins normally inhibit MAPK signaling in the hub. A.U., arbitrary units. Black bars indicate the mean and standard error. Unpaired t test, **p < 0.01, ****p < 0.0001. (**E–F**) Single confocal sections through the apex of testes after 8 days of G-TRACE lineage tracing system expression in adult hub cells. Testes were immunostained for RFP (red, current expression of the hub-specific driver *E132-Gal4*) and GFP (green, current and past expression of the driver) and counterstained with DAPI (blue; marks all nuclei). Hubs are outlined in white. (**E'-F'**) Red channel alone, in white; (**E"-F"**) green channel alone, in white. After knockdown of Sprouty (**E**) or PTEN (**F**) in the hub together with expression of G-TRACE, hub cells are marked with RFP and GFP, and cells outside the hub are also marked, either with GFP and low levels of RFP (yellow arrowheads) or with GFP only (white arrowheads), suggesting that marked cyst lineage cells arose by conversion of hub cells to CySCs. Scale bars in D (for A-B, D) and in F (for E-F) are 10 μm.

## EGFR signaling is upregulated in hub cells in testes recovering from CySC ablation

Having determined that inhibitors of EGFR signaling maintain adult hub cell quiescence and that EGFR pathway activation can cause hub cells to re-enter the cell cycle and transdifferentiate into cyst lineage cells, we asked if EGFR signaling plays a role in testes recovering from CySC ablation, a condition which also stimulated hub cell transdifferentiation (*Hétié et al., 2014*). We began by asking if we could detect activation of the EGFR pathway in hub cells in these testes. Genetic ablation of CySCs is accomplished by using the cyst lineage driver *C587-Gal4, Gal80ts* to drive expression of the pro-apoptotic gene *grim* in adult testes (*C587-Gal4, Gal80ts > UAS* grim); after *grim* expression is withdrawn, hub cells in most testes re-enter the cell cycle, eventually giving rise to new CySCs (*Hétié et al., 2014*). To assay for EGFR pathway activation, we again used dpERK as an indicator. We shifted control (*C587-Gal4, Gal80ts*) and experimental (*C587-Gal4, Gal80ts > UAS* grim) flies to restrictive

**Table 4.** Overexpression of EGF ligands in the adult testis niche does not cause hub cell proliferation.

| Gal4 driver | UAS line | Source | Days at 31 ℃ | % Testes with PH3-positive hub cells |
|---|---|---|---|---|
| C587-Gal4, Gal80ts | UAS-GFP-RNAi | BDSC 9330 or 9331 | 7 | 0% (n = 0/294)* |
| | UAS-secreted spitz | BDSC 63134 | 7–9 | 0% (n = 0/219)[a,ns] |
| | UAS-secreted spitz | BDSC 58436 | 7 | 0% (n = 0/75)[a,ns] |
| | UAS-gurken ΔTC | *Queenan et al., 1999* | 7 | 0% (n = 0/82)[a,ns] |
| | UAS-secreted gurken | BDSC 58417 | 7 | 0% (n = 0/118)[a,ns] |
| | UAS-secreted keren | *Urban et al., 2002* | 7 | 0% (n = 0/113)[a,ns] |
| | UAS-vein | *Schnepp et al., 1996* | 6 | 0% (n = 0/96)[a,ns] |
| E132-Gal4, Gal80ts | UAS-GFP-RNAi | BDSC 9330 or 9331 | 7 | 0% (n = 0/237)* |
| | UAS-secreted spitz | BDSC 63134 | 7 | 0% (n = 0/208)[a,ns] |
| | UAS-secreted spitz | BDSC 58436 | 7 | 0% (n = 0/92)[a,ns] |
| | UAS-gurken ΔTC | *Queenan et al., 1999* | 7 | 0% (n = 0/86)[a,ns] |
| | UAS-secreted gurken | BDSC 58417 | 7 | 1% (n = 1/77)[a,ns] |
| | UAS-secreted keren | *Urban et al., 2002* | 7 | 0% (n = 0/113)[a,ns] |
| | UAS-vein | *Schnepp et al., 1996* | --- | --- |
| C587-Gal4, Gal80ts | UAS-secreted spitz +UAS-Argos-RNAi | BDSC 63134 + BDSC 28383 | 7–9 | 2% (n = 4/203)[b,ns] |

Fisher's Exact Test: ns = not significant (a, compared to UAS-GFP-RNAi control; b, compared to UAS-Argos-RNAi BDSC 28383 alone [*Table 1*]).
*Data from *Table 1*.

temperature (31 °C) for 2 days to ablate all early cyst lineage cells in the experimental flies, and then returned them to permissive temperature (18 °C) for 2 days. As expected based on previous studies, in control testes, immunostaining for dpERK revealed consistently high pathway activation in cyst lineage cells but not in hub cells (*Figure 4A*; *Chen et al., 2013*; *Fairchild et al., 2016*; *Kiger et al., 2000*). By contrast, in experimental testes, the levels of dpERK staining in the hub were significantly higher than in control testes (*Figure 4B–C*), suggesting that the EGFR pathway activity is up-regulated in hub cells after CySC ablation. As expected, dpERK staining levels appear much lower in the area adjacent to the hub in cyst lineage-ablated testes compared to controls, since most testes have not yet recovered early cyst lineage cells at 2 days post-ablation (*Hétié et al., 2014*).

Given that EGFR activation increases in hub cells as they lose quiescence following CySC ablation, we next asked if this pathway is important for transdifferentiation of hub cells into CySCs. Tools to knock down gene expression in the hub in testes recovering from CySC ablation do not yet exist; we therefore addressed this question using a gene dosage approach. We genetically ablated CySCs and early cyst cells using *C587-Gal4, Gal80^{ts}* to drive expression of *grim* as before (*Hétié et al., 2014*) in otherwise normal control flies (*Egfr+/+*) and also in flies null for one copy of the *Egfr* gene (*Egfr-/+*). In both genotypes, testes were indistinguishable from wild type testes before ablation (data not shown), and after ablation, most testes lacked CySCs and early cyst cells but retained hub cells and germ cells as expected (*Figure 4D*; *Table 5*; *Hétié et al., 2014*, *Lim and Fuller, 2012*). After 7 days of recovery, consistent with previous studies, 80% of control *Egfr+/+* testes (n = 225) had regained CySCs and early cyst cells while maintaining a hub and germ cells. Most of the remaining 20% of testes had only a hub or a hub and germ cells at their apex (*Figure 4D–I*; *Table 5*). By contrast, testes from *Egfr-/+* flies had a significantly different distribution of phenotypes, with only a minority of testes (45%, n = 142) regaining CySCs and early cyst cells while maintaining a hub and germ cells; most testes had only a hub or a hub and germ cells at their apex (*Figure 4D*; *Table 5*). Fourteen days after CySC ablation, the percentage of testes that had regained CySCs and early cyst cells while maintaining a hub and germ cells remained lower in *Egfr-/+* flies (55%, n = 116) than in control *Egfr+/+* flies (81%, n = 151; *Table 5*). These data are consistent with the hypothesis that EGFR signaling is important for the transdifferentiation of hub cells into CySCs after ablation.

## EGFR signaling promotes the formation of ectopic niches in testes recovering from CySC ablation

Ectopic niches (additional hubs that support GSCs and CySCs) can form de novo in adult testes either after extended recovery from CySC ablation, or after hub cells are forced to re-enter the cell cycle by knockdown of Rbf or over-expression of cell cycle activators (*Greenspan and Matunis, 2018*; *Hétié et al., 2014*). To ask if EGFR signaling is required for ectopic niche formation after extended recovery from CySC ablation (*C587-Gal4, Gal80^{ts}ts grim*), we compared the number of testes with ectopic niches in *Egfr-/+* flies and control *Egfr+/+* flies after 14 days of recovery. We determined the percentage of testes with ectopic niches out of the total number of testes that had recovered their cyst lineage cells, since ectopic niches have been found only in this population of testes. The percentage of recovered testes with ectopic niches in *Egfr-/+* flies (12%, n = 270) was significantly lower than in control *Egfr+/+* flies (29%, n = 487, p < 0.0001; *Table 6*). We conclude that EGFR signaling is important not only for recovery of CySCs but also for ectopic niche formation in testes recovering from genetic ablation of CySCs.

We then asked if activation of EGFR signaling alone, in otherwise unperturbed testes, can cause ectopic niches to form after an extended period of time (14 days). We drove expression of constitutively active EGFR or Sprouty RNAi in adult hub cells (using *E132-Gal4, Gal80^{ts}*) or Argos RNAi in adult cyst lineage cells (using *C587-Gal4, Gal80^{ts}*), but we found no ectopic niches in any testes (n = 35, 285, or 291 testes, respectively). Therefore, although EGFR signaling is important for ectopic niche formation after genetic ablation of CySCs, activation of EGFR signaling by itself is not sufficient to promote the formation of ectopic niches.

## Discussion

Here, we show that activation of the EGFR signaling pathway in the hub is sufficient to drive hub cell proliferation and conversion to cyst lineage cells in adult testes. Moreover, this pathway must be

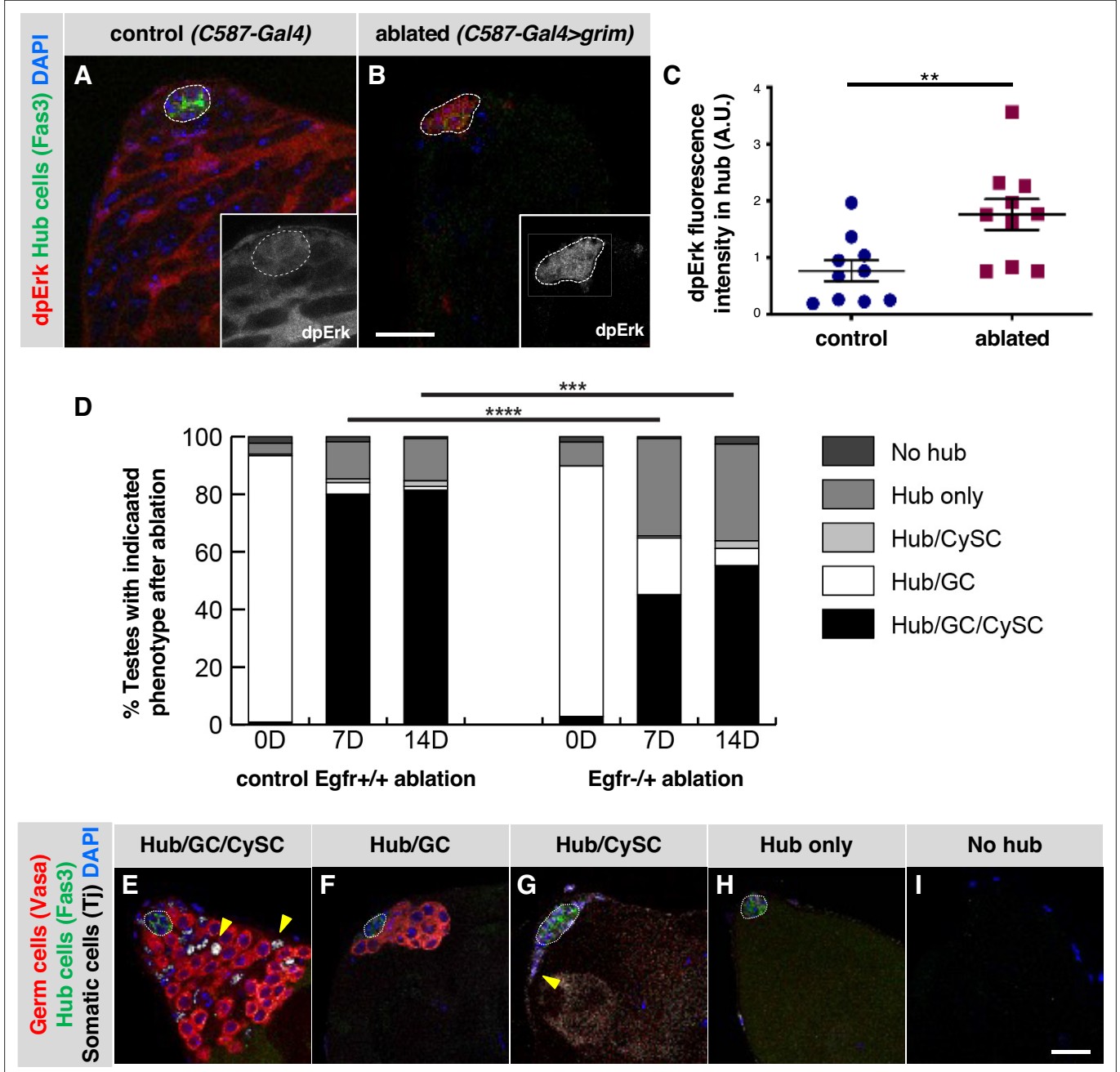

**Figure 4.** EGFR signaling is important for testis recovery from CySC ablation. (**A–B**) Single confocal sections through the apex of testes immunostained for Fas3 (hub cell membranes, green), dpERK (EGFR pathway activation, red), and counterstained with DAPI (nuclei, blue). Hubs are outlined in white. Insets show the red channel alone, in white and enlarged. In control *C587-Gal4, Gal80^ts* testes (**A**), dpERK levels are high in cyst lineage cells (indicating high levels of EGFR pathway activation) but low in hub cells. In *C587-Gal4, Gal80^ts > UAS grim* testes (**B**), 2 days after genetic ablation of all CySCs and early cyst cells, dpERK levels are high in hub cells. Scale bar in B (for A-B) is 20 μm. (**C**) Quantification of dpERK levels in the hub in control (**A**) and ablated (**B**) testes. dpErk levels in the hub are significantly higher in ablated testes than in control testes. A.U., arbitrary units. Black bars indicate the mean and standard error. Unpaired t test, \*\*p < 0.01. (**D**) Bar graph showing the distribution of testis phenotypes in control *C587-Gal4, Gal80^ts > UAS grim, Egfr+/+* flies ("control Egfr+/+ ablation") and in *C587-Gal4, Gal80^ts > UAS grim, Egfr-/+* flies ("EGFR-/+ ablation") at 0, 7, or 14 days after genetic ablation of CySCs and early cyst cells. After ablation (0 days), in both control and *Egfr-/+* flies, most testes lack all CySCs and early cyst cells but retain a hub and germ cells (GC) as expected (white bars). At 7 and 14 days after ablation, fewer testes have regained CySCs and early cyst cells (black bars) in *Egfr-/+* flies than in control flies and there is a significant difference in phenotype distribution. Chi square test, \*\*\*p < 0.001, \*\*\*\*p < 0.0001. (**E–I**) Single confocal sections through the apex of testes at 7 days after ablation, immunostained for Vasa (germ cells, red), Fas3 (hub cell membranes, green), and Tj (somatic cell nuclei, white), and counterstained with DAPI (nuclei, blue), to illustrate the phenotypes listed in (**D**). Testes that recover CySCs and early cyst lineage cells have Tj-positive nuclei outside the hub (yellow arrowheads); most also contain germ cells (**E**) but a few contain just a hub and

*Figure 4 continued on next page*

*Figure 4 continued*

cyst cells (**G**). Testes that fail to recover CySCs and early cyst lineage cells can retain a hub and germ cells (**F**) or just a hub (**H**) or no hub or germ cells (**I**). Hubs are outlined in white. Scale bar in I (for E-I) is 20 μm.

actively inhibited in adult hub cells to maintain their quiescence and fate, since loss of the pathway inhibitors Sprouty, Argos, or PTEN is sufficient to drive these phenotypes. We also show that the EGFR pathway is activated in hub cells upon genetic ablation of CySCs and is important for recovery of CySCs from hub cells after ablation. Signaling pathways other than EGFR could also be involved in driving hub cells out of quiescence after CySC ablation.

In adult testes, the EGF ligand Spitz is known to be secreted from germ cells and received by cyst cells, where activation of the EGFR pathway is essential for their proper encystment of the germ cells (*Sarkar et al., 2007*; *Schulz et al., 2002*). In the absence of cyst lineage cells – for example, after genetic ablation of CySCs and early cyst cells – we propose that EGF ligands are still secreted from the germ cells but are received by the hub cells instead, promoting their conversion to new CySCs. In support of this model, after genetic ablation of all CySCs and early cyst lineage cells, testes rarely recover CySCs in the absence of germ cells (*Figure 4*; *Table 5*; *Hétié et al., 2014*). Furthermore, hub cells do not divide and convert to CySCs after genetic ablation of only some, but not all, CySCs and early cyst lineage cells (*Hétié et al., 2014*), consistent with the idea that hub cell quiescence is maintained by signals from cyst lineage cells. We propose a model (*Figure 5*) where, after CySC ablation, hub cell proliferation and conversion to CySCs are driven by EGFR signaling, promoted by a new signal (EGF ligands from germ cells) not normally received by the hub, and by loss of an existing signal (the EGFR inhibitor Argos from cyst lineage cells). Once CySCs recover, the inhibitory signal returns, causing EGFR signaling levels in the hub to diminish. Recovery from genetically-induced tissue damage thus depends on the coordination of signals between different cell types within the testis stem cell niche. Future studies of the cellular distribution of EGFR signaling pathway components within the niche could be informative in understanding how these signals are coordinated in unperturbed and recovering niches.

A similar example of cellular plasticity in a stem cell niche was recently described in the *Drosophila* ovary, where two types of stem cells, germline stem cells and somatic follicle stem cells (FSCs), are housed in distinct niches at the apical ends of each ovary. A niche for FSCs is created by adjacent escort cells (aka inner germarial sheath cells), which do not give rise to new FSCs under normal conditions; however, under starvation conditions or upon forced activation of mTOR or Toll signaling in escort cells, these cells can convert into new FSCs (*Rust et al., 2020*). Another example comes from the mouse small intestine, where quiescent Paneth cells, which create a niche for adjacent *Lgr5+* stem cells, can replace stem cells that are lost as a result of inflammation-induced damage or

**Table 5.** Ablation phenotypes with and without reduced EGFR.

| Genotype | Days recovered | Hub/GC/CySC | Hub/GC | Hub/CySC | Hub only | No hub |
|---|---|---|---|---|---|---|
| C587-Gal4; UAS-Grim/+; Tub-Gal80[ts] | 0 | < 1% (n = 2/213) | 93% (n = 197/213) | < 1% (n = 1/213) | 4% (n = 8/213) | 2% (n = 5/213) |
| C587-Gal4; UAS-Grim/ Egfr[F24]; Tub-Gal80[ts] | 0[*,§,ns] | 3% (n = 3/108) | 87% (n = 94/108) | 0% (n = 0/108) | 8% (n = 9/108) | 2% (n = 2/108) |
| C587-Gal4; UAS-Grim/+; Tub-Gal80[ts] | 7 | 80% (n = 180/225) | 4% (n = 9/225) | 1% (n = 3/180) | 13% (n = 29/225) | 2% (n = 4/225) |
| C587-Gal4; UAS-Grim/ Egfr[F24]; Tub-Gal80[ts] | 7[†,§,****] | 45% (n = 64/142) | 20% (n = 28/142) | < 1% (n = 1/142) | 34% (n = 48/142) | < 1% (n = 1/142) |
| C587-Gal4; UAS-Grim/+; Tub-Gal80[ts] | 14 | 81% (n = 123/151) | 1% (n = 2/151) | 2% (n = 3/151) | 15% (n = 22/151) | < 1% (n = 1/151) |
| C587-Gal4; UAS-Grim/ Egfr[F24]; Tub-Gal80[ts] | 14[‡,§,***] | 55% (n = 64/116) | 6% (n = 7/116) | 3% (n = 3/116) | 33% (n = 39/116) | 3% (n = 3/116) |

[*]Compared with C587-Gal4; UAS-Grim/+; Tub Gal80[ts] at 0 days recovered.
[†]Compared with C587-Gal4; UAS-Grim/+; Tub Gal80[ts] at 7 days recovered.
[‡]Compared with C587-Gal4; UAS-Grim/+; Tub Gal80[ts] at 14 days recovered.
[§]Chi Square Test: ***$p < 0.001$, ****$p < 0.0001$, ns = not significant.

**Table 6.** Percentage of testes with ectopic hubs after 14 day recovery from CySC ablation.

| Genotype | Days Recovered | % Recovered Testes with Ectopic Hubs | % Recovered Testes without Ectopic Hubs |
|---|---|---|---|
| C587-Gal4; UAS-Grim/+; Tub-Gal80$^{ts}$ | 14 | 29% (n = 139/487) | 71% (n = 348/487) |
| C587-Gal4; UAS-Grim/Egfr$^{F24}$; Tub-Gal80$^{ts}$ | 14**** | 12% (n = 32/270) | 88% (n = 238/270) |

Fisher's Exact Test: ****$p < 0.0001$ (compared to testes with ectopic hubs in control C587-Gal4; UAS-Grim/+; Tub-Gal80$^{ts}$ flies).

irradiation to the tissue (*Schmitt et al., 2018*; *Yu et al., 2018*). The ability of quiescent niche cells to convert to stem cells in response to tissue damage is therefore likely to be a general feature of adult stem cell niches.

In flies recovering from genetic ablation of CySCs, when *Egfr* gene dosage is reduced, transdifferentiation of hub cells into CySCs and ectopic niche formation are both affected, happening less often than in control flies (*Figure 4*; *Table 5*; *Table 6*). However, in otherwise wild-type flies, forced activation of EGFR signaling in hub cells is sufficient for hub cell transdifferentiation (*Table 2*) but not for ectopic niche formation. This result is surprising because hub cell transdifferentiation and ectopic niche formation are both common in testes with forced expression of cell cycle activators or knockdown of a cell cycle inhibitor in hub cells (*Hétié et al., 2014*; *Greenspan and Matunis, 2018*). Interestingly, no ectopic niches were reported in testes after depletion of the transcription factor Escargot from the hub, which causes hub cells to transdifferentiate into CySCs without also re-entering the cell cycle, resulting in the complete loss of hub cells over time (*Voog et al., 2014*). Taken together, these studies suggest that the transdifferentiation of hub cells to CySCs does not by itself cause ectopic niche formation, and that additional factors must come into play in some cases to push testes towards this abnormal phenotype. This is of interest, since the regulation of niche number across tissues, particularly during regeneration, is not understood. What factors regulate regeneration, and how multiple signals from different cell types interact after injury to a tissue to ensure its proper recovery, are questions with important implications for regenerative medicine.

## Materials and methods

### *Drosophila* husbandry and strains

Flies were raised on a standard yeast/cornmeal/molasses medium (1212.5 mL water, 14.7 mL agar, 20.4 g yeast, 81.8 g cornmeal, 109.1 ml molasses, 10.9 mL Tegosept, 3.4 mL propionic acid, 0.4 mL phosphoric acid per tray of 100 vials) supplemented with dry yeast at 18 °C unless otherwise indicated. Male flies 0–7 days old were used for all experiments and subjected to different conditions as noted within the text, figure legends, and methods. Flies containing the UAS-Grim construct (*Wing et al., 1998*) were a gift from DJ Pan; UAS-gurken ΔTC (*Queenan et al., 1999*) and UAS-secreted keren (*Urban et al., 2002*) flies were a gift from D. Montell; and UAS-vein flies (*Schnepp et al., 1996*) were a gift from S. Hou. All other stocks can be obtained from the Bloomington *Drosophila* Stock Center (BDSC) or Vienna *Drosophila* Resource Center (VDRC). See main tables and Key Resources Table for list of strains used for each experiment. See Screen Summary Table for additional lines analyzed.

### Transgene induction

Flies containing UAS and Gal4 constructs together with tub-Gal80$^{ts}$ were grown to adulthood at permissive temperature (18 °C) and shifted to non-permissive temperature (29 °C or 31 °C, as indicated) to induce expression of UAS transgenes (RNAi, dominant-negative, or over-expression constructs).

### CySC ablation (*Hétié et al., 2014*)

Flies containing C587-Gal4, UAS-grim, and tub-Gal80$^{ts}$ transgenes were grown at 18 °C and shifted to 31 °C for 2 days to induce cell death in all CySCs and early cyst lineage cells. Flies were then dissected or returned to 18 °C to recover for 7 or 14 days as indicated. In every experiment with C587-Gal4, Gal80$^{ts}$>UAS-grim, 3–5% of testes look completely normal, like unperturbed wild-type testes, after

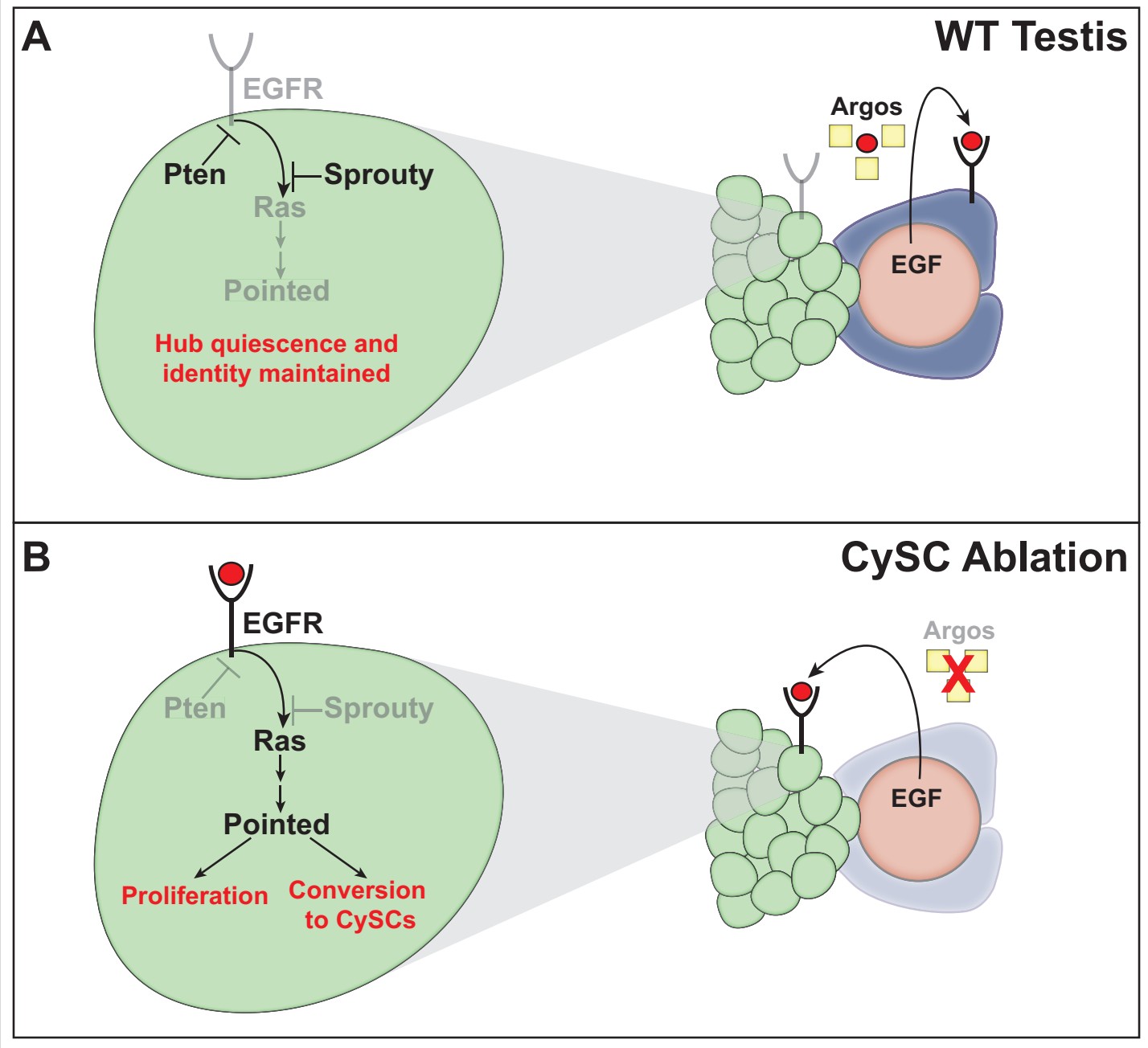

**Figure 5.** Model for cyst lineage recovery after ablation. (**A**) In wild type testes, EGF ligands (red circles) are secreted by germ cells and received by cyst lineage cells. The EGFR pathway is repressed in hub cells by the secreted inhibitor Argos (yellow squares), which sequesters EGF ligands, and by intrinsic pathway inhibitors (Sprouty and PTEN). Hub quiescence and identity are maintained. (**B**) After genetic ablation of all CySCs and early cyst lineage cells, EGF ligands are no longer sequestered by Argos and are received by the hub. The EGFR pathway is activated in hub cells, driving expression of the downstream transcription factor Pointed and its target genes, resulting in hub cell proliferation and conversion to CySCs. As new CySCs are generated, Argos is once again expressed, down-regulating EGFR signaling in the hub.

the shift to 31 °C; we speculate that a transgene has been lost from these flies. These 'unablated' testes remain distinguishable from 'ablated' testes throughout the experiment and are not included in our results or statistical analyses.

## Dissection and immunohistochemistry (*Matunis et al., 1997*)

Flies were anesthetized using $CO_2$ and dissected in 1 X Ringer's solution (111 mM NaCl, 1.88 mM KCl, 64 µM $NaH_2PO_4$, 816 µM $CaCl_2$, 2.38 mM $NaHCO_3$; *Ashburner, 1989*). All steps were performed at room temperature unless otherwise noted. Testes with attached cuticle were transferred to fixation solution (4% paraformaldehyde in 1 X PBS with 0.1% Trition X-100) and placed on a rocking platform for 22 min.

After fixation, testes were washed in 1 X PBX (1 X PBS with 0.1% Trition X-100) and blocked for one hour at room temperature or overnight at 4 °C in 1 X PBX plus 3% BSA, 0.02% $NaN_3$, and 2% goat or donkey serum. Primary and secondary antisera were diluted into 1 X PBX with 3% BSA and 0.02% $NaN_3$. Testes were incubated in primary antisera overnight at 4 °C on a rocking platform, washed in 1 X PBX, incubated in secondary antisera for two hours at room temperature or overnight at 4 °C, and again washed in 1 X PBX. The nuclear counterstain 4,6-diamidino-2-phenylindole (DAPI; Millapore/Sigma) was added to secondary antisera solutions at a final concentration of 1 µg/mL. After the final wash step, testes were rinsed in 1 X PBS and transferred to Vectashield antifade mounting medium (Vector Laboratories).

For dpErk staining experiments, testes were dissected in Shields and Sang M3 media with 1:100 dilution of phosphatase inhibitor cocktail 2, in place of Ringer's solution, and washed in 1 X Testis Buffer with phosphatase inhibitor, in place of PBX (*Fairchild et al., 2016*).

All polyclonal antisera and mouse anti-Phospho-Histone H3 were diluted 1:1 in glycerol and stored at –20 °C; other monoclonal antibodies were stored at 4 °C. Antisera were used at the following final concentrations: mouse anti-Fasciclin III 7G10 (1:50; Developmental Studies Hybridoma Bank), mouse anti-Phospho-Histone H3 (1:400; Cell Signaling Technology), guinea pig anti-Traffic Jam (1:20,000; a gift from D. Godt), rabbit anti-Vasa (1:200; Santa Cruz Biotechnology), chick anti-GFP (1:10,000; Abcam), rabbit anti-dsRed (1:10,000; Takara Bio), rabbit anti-dpErk (1:100, Cell Signaling Technology). Fluor 488 secondary antibodies were used at a final concentration of 1:400; other secondary antibodies were used at a final concentration of 1:200.

## Lineage tracing

Driving expression of the G-TRACE construct in the hub (E132-Gal4, Gal80[ts]>UAS-G-TRACE) caused permanent expression of GFP in hub cells and their descendants. Marked cells were detected by immunostaining for GFP and RFP. Testes with GFP-marked cells outside the confines of the hub cluster that no longer expressed RFP were considered positive for converting cells. The hub cluster was defined as tightly packed cells marked by bright RFP.

## Microscopy and image analysis

Fixed images were obtained using a Zeiss LSM Pascal equipped with a 63 x oil immersion objective, 405 nm diode, 488 nm ArKr, and 543 nm HeNe lasers with digital zoom; a Zeiss LSM 700 (JHU SOM microscope facility) equipped with a 63 x oil immersion objective, 405 nm diode, 488 nm solid-state, 561 nm solid-state, and 639 nm diode lasers with digital zoom; or a Zeiss LSM 800 equipped with a 63 x oil immersion objective, 405 nm, 488 nm, 561 nm, and 640 nm diode lasers with digital zoom and GaAsP detectors. Images were acquired using the Zeiss LSM or Zen software and processed using Zeiss LSM, Zen, or Fiji. Brightness for individual channels from single confocal slices was enhanced using Fiji, and then the channels were overlaid to form a merged image.

## Quantification of dividing hub cells

To quantify hub cell divisions, testes were immunostained with the mitotic marker Phospho-Histone H3. Testes with PH3-positive cells within the confines of the hub cluster were considered positive for dividing hub cells. The hub cluster was defined as those cells marked by the hub membrane marker Fas3.

## dpErk fluorescence intensity measurements

To measure levels of dpErk expression, testes were immunostained with dpErk, Fas3, and DAPI. Z stacks with a 0.5 µm step size were acquired to include the entire hub range. Fluorescence signal was

acquired at the same gain in the linear range for all samples. Fluorescence intensity was measured using FIJI software. All stacks containing the hub, as indicated by Fas3 staining, were merged into a single summed slice and dpErk fluorescence intensity was measured by drawing an object around the entire hub and using the measure feature. The corrected total cell fluorescence (CTCF) was then calculated by taking the integrated density and subtracting the area times the mean of 3 fluorescence background readings. The CTCF for dpErk fluorescence was then normalized over the CTCF for the DAPI channel, which was calculated in the same manner. The normalized dpErk fluorescence measurements in the hub were compared for testes whose CySCs were ablated to those that had not undergone ablation.

### Ectopic niches quantification

To quantify ectopic niches, testes were immunostained with the hub membrane marker Fasciclin 3. Z stacks were acquired to include the entire hub range. Clusters of hub cells whose membranes were no longer connected to other clusters in any Z planes were considered separate hubs. Testes with more than one hub cluster surrounded by germline and somatic cells were considered positive for ectopic niches.

### Statistical analysis

For all quantifications, $n$ represents the number of testes analyzed. Statistical significance was expressed as p values and determined using a Fisher's exact test for most measurements except ablation distribution phenotypes, in which Chi-square test was used, and dpErk fluorescence measurements, in which an unpaired t test was used. All statistical tests were run using PRISM 9 software. (*) denotes $p < 0.05$, (**) denotes $p < 0.01$, (***) denotes $p < 0.001$, (****) denotes $p < 0.0001$, and (ns) denotes values that were not significant.

## Acknowledgements

We thank the "Screen Team" (Zelalem Demere, Jean Garcia, Linh Pham, Eli Ross, Jhanavi Sivakumar, and Tiffaney Tran) for countless hours of dissecting, mounting, and imaging fly testes; Alyshia Scholl for help with building fly stocks; DJ Pan, Denise Montell, the Bloomington *Drosophila* Stock Center, and the Vienna *Drosophila* Resource Center for fly stocks; Dorothea Godt and Developmental Studies Hybridoma Bank for antibodies; and Mara Grace for comments on the manuscript. This work was funded by NIH grants R01HD052937, R35GM136665 (to ELM), and F31HD085748 (to LJG).

## Additional information

### Funding

| Funder | Grant reference number | Author |
| --- | --- | --- |
| Eunice Kennedy Shriver National Institute of Child Health and Human Development | F31HD085748 | Leah J Greenspan |
| Eunice Kennedy Shriver National Institute of Child Health and Human Development | R01HD052937 | Erika L Matunis |
| National Institute of General Medical Sciences | R35GM136665 | Erika L Matunis |

The funders had no role in study design, data collection and interpretation, or the decision to submit the work for publication.

### Author contributions

Leah J Greenspan, Conceptualization, Data curation, Formal analysis, Funding acquisition, Investigation, Methodology, Project administration, Supervision, Validation, Visualization, Writing – original

draft, Writing – review and editing; Margaret de Cuevas, Conceptualization, Data curation, Formal analysis, Investigation, Methodology, Project administration, Supervision, Validation, Visualization, Writing – original draft, Writing – review and editing; Kathy H Le, Formal analysis, Investigation, Methodology; Jennifer M Viveiros, Formal analysis, Investigation, Writing – review and editing; Erika L Matunis, Conceptualization, Funding acquisition, Methodology, Project administration, Supervision, Writing – original draft, Writing – review and editing

## Author ORCIDs
Leah J Greenspan (ID) http://orcid.org/0000-0001-6368-8007
Margaret de Cuevas (ID) http://orcid.org/0000-0002-9401-968X

## Decision letter and Author response
Decision letter https://doi.org/10.7554/eLife.70810.sa1
Author response https://doi.org/10.7554/eLife.70810.sa2

## Additional files

### Supplementary files
• Transparent reporting form

### Data availability
All data generated or analyzed during this study are included in the manuscript.

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

# Appendix 1

## Appendix 1—key resources table

| Reagent type (species) or resource | Designation | Source or reference | Identifiers | Additional information |
|---|---|---|---|---|
| Genetic reagent (*D. melanogaster*) | E132Gal4: P{w[ + mW.hs] = GawB}E132, w[*] (also called upd-Gal4) | Bloomington *Drosophila* Stock Center | RRID:BDSC_26796 | |
| Genetic reagent (*D. melanogaster*) | TubGal80[ts]: w[*];P{w[ + mC] = tubP-GAL80[ts]}2/TM2 | Bloomington *Drosophila* Stock Center | RRID:DSC_7017 | |
| Genetic reagent (*D. melanogaster*) | UAS-GFP RNAi: w[1118]; P{w[ + mC] = UAS GFP. dsRNA.R}142 | Bloomington *Drosophila* Stock Center | RRID:BDSC_9330 | |
| Genetic reagent (*D. melanogaster*) | UAS-GFP RNAi: w[1118]; P{w[ + mC] = UAS GFP. dsRNA.R}143 | Bloomington *Drosophila* Stock Center | RRID:BDSC_9331 | |
| Genetic reagent (*D. melanogaster*) | UAS-Rbf RNAi: y[1] sc[*] v[1]; P{y[ + t7.7] v[ + t1.8] = TRiP. HMS03004}attP2/TM3, Sb[1] | Bloomington *Drosophila* Stock Center | RRID:BDSC_36744 | |
| Genetic reagent (*D. melanogaster*) | UAS-Rbf RNAi: y[1] sc[*] v[1]; P{y[ + t7.7] v[ + t1.8] = TRiP . GL01293}attP40/CyO | Bloomington *Drosophila* Stock Center | RRID:BDSC_41863 | |
| Genetic reagent (*D. melanogaster*) | C587Gal4: P{w[ + mW.hs] = GawB}C587, w[*] | Bloomington *Drosophila* Stock Center | RRID:BDSC_67747 | |
| Genetic reagent (*D. melanogaster*) | UAS-Pointed.P1: w[1118]; P{w[ + mC] pnt[P1.UAS] = UAS pnt.P1}3 | Bloomington *Drosophila* Stock Center | RRID:BDSC_869 | |
| Genetic reagent (*D. melanogaster*) | UAS-Pointed.P2: w[1118]; P{w[ + mC] pnt[P2.UAS] = UAS pnt.P2}2/TM3, Sb[1] | Bloomington *Drosophila* Stock Center | RRID:BDSC_399 | |
| Genetic reagent (*D. melanogaster*) | UAS-Ras85D.V12: w[1118]; P{w[ + mC] = UAS-Ras85D. V12}TL1 | Bloomington *Drosophila* Stock Center | RRID:BDSC_4847 | |
| Genetic reagent (*D. melanogaster*) | UAS-Rolled: y[1] w[*]; P{w[ + mC] = UAS-rl[Sem].S}2 | Bloomington *Drosophila* Stock Center | RRID:BDSC_59006 | |
| Genetic reagent (*D. melanogaster*) | UAS-Egfr Type I: w[*]; P{w[ + mC] = Egfr.1 .A887T. UAS}12–4/CyO, P{ry[ + t7.2] = sevRas1 .V12}FK1 | Bloomington *Drosophila* Stock Center | RRID:BDSC_9534 | |
| Genetic reagent (*D. melanogaster*) | UAS-Egfr Type II: w[*]; P{w[ + mC] = Egfr.2 .A887T. UAS}8–2 | Bloomington *Drosophila* Stock Center | RRID:BDSC_9533 | |
| Genetic reagent (*D. melanogaster*) | UAS-Egfr $\lambda$ : w[*]; P{w[ + mC] = UAS Egfr.lambdatop}3/ TM6C, Sb[1] | Bloomington *Drosophila* Stock Center | RRID:BDSC_59843 | |
| Genetic reagent (*D. melanogaster*) | UAS-PVR $\lambda$ : w[1118]; P{w[ + mC] = UASp Pvr.lambda} mP10 | Bloomington *Drosophila* Stock Center | RRID:BDSC_58496 | |
| Genetic reagent (*D. melanogaster*) | UAS-PVR $\lambda$ : w[1118]; P{w[ + mC] = UASp Pvr.lambda} mP1 | Bloomington *Drosophila* Stock Center | RRID:BDSC_58428 | |
| Genetic reagent (*D. melanogaster*) | UAS-InR: y[1] w[1118]; P{w[ + mC] = UAS InR.K414P}2 | Bloomington *Drosophila* Stock Center | RRID:BDSC_8250 | |
| Genetic reagent (*D. melanogaster*) | UAS-InR: y[1] w[1118]; P{w[ + mC] = UAS InR.A1325D}2 | Bloomington *Drosophila* Stock Center | RRID:BDSC_8263 | |
| Genetic reagent (*D. melanogaster*) | UAS-Heartless $\lambda$ : y[1] w[*]; P{w[ + mC] = UAS htl. lambda.M}40-22-2 | Bloomington *Drosophila* Stock Center | RRID:BDSC_5367 | |
| Genetic reagent (*D. melanogaster*) | UAS-Breathless $\lambda$ : w[*]; P{w[ + mC] = UAS btl. lambda}2 | Bloomington *Drosophila* Stock Center | RRID:BDSC_29045 | |

*Appendix 1 Continued on next page*

*Appendix 1 Continued*

| Reagent type (species) or resource | Designation | Source or reference | Identifiers | Additional information |
|---|---|---|---|---|
| Genetic reagent (*D. melanogaster*) | UAS-Sprouty RNAi: y[1] sc[*] v[1] sev[21]; P{y[ + t7.7] v[ + t1.8] = TRiP.HMS01599}attP2 | Bloomington *Drosophila* Stock Center | RRID:BDSC_36709 | |
| Genetic reagent (*D. melanogaster*) | UAS-Pten RNAi: y[1] v[1]; P{y[ + t7.7] v[ + t1.8] = TRiP.HMS00044}attP2 | Bloomington *Drosophila* Stock Center | RRID:BDSC_33643 | |
| Genetic reagent (*D. melanogaster*) | UAS-Argos RNAi: y[1] v[1]; P{y[ + t7.7] v[ + t1.8] = TRiP.JF03020}attP2 | Bloomington *Drosophila* Stock Center | RRID:BDSC_28383 | |
| Genetic reagent (*D. melanogaster*) | UAS-Egfr DN: y[1] w[*]; P{w[ + mC] = UAS Egfr.DN.B}29-77-1; P{w[ + mC] = UAS Egfr.DN.B}29-8-1 | Bloomington *Drosophila* Stock Center | RRID:BDSC_5364 | |
| Genetic reagent (*D. melanogaster*) | UAS-Egfr RNAi: y[1] v[1]; P{y[ + t7.7] v[ + t1.8] = TRiP.JF02283}attP2 | Bloomington *Drosophila* Stock Center | RRID:BDSC_36770 | |
| Genetic reagent (*D. melanogaster*) | UAS-Egfr RNAi: y[1] v[1]; P{y[ + t7.7] v[ + t1.8] = TRiP.HMS05003}attP40 | Bloomington *Drosophila* Stock Center | RRID:BDSC_60012 | |
| Genetic reagent (*D. melanogaster*) | UAS-Egfr RNAi: w[1118]; P{GD1654}v43267 | Vienna *Drosophila* Resource Center | Stock #: 43,267 | |
| Genetic reagent (*D. melanogaster*) | UAS-Egfr RNAi: P{KK100051}VIE-260B | Vienna *Drosophila* Resource Center | Stock #: 107,130 | |
| Genetic reagent (*D. melanogaster*) | UAS-N-CA: P{ry[ + t7.2] = hsFLP}1, y[1] w[*]; P{w[ + mC] = UAS N.intra.GS}2/CyO; MKRS/TM2 | Bloomington *Drosophila* Stock Center | RRID:BDSC_52008 | No longer available |
| Genetic reagent (*D. melanogaster*) | UAS-N-DN: y[ + t7.2] = hsFLP}12, y[1] w[*]; P{w[ + mC] = UAS N.ECN}2; MKRS/TM2 | Bloomington *Drosophila* Stock Center | RRID:BDSC_51667 | |
| Genetic reagent (*D. melanogaster*) | UAS-Notch RNAi: y[1] v[1]; P{y[ + t7.7] v[ + t1.8] = TRiP.HMS00001}attP2 | Bloomington *Drosophila* Stock Center | RRID:BDSC_33611 | |
| Genetic reagent (*D. melanogaster*) | UAS-Notch RNAi: y[1] v[1]; P{y[ + t7.7] v[ + t1.8] = TRiP.HMS00009}attP2 | Bloomington *Drosophila* Stock Center | RRID:BDSC_33616 | |
| Genetic reagent (*D. melanogaster*) | UAS-Notch RNAi: y[1] sc[*] v[1] sev[21]; P{y[ + t7.7] v[ + t1.8] = TRiP .GL00092}attP2 | Bloomington *Drosophila* Stock Center | RRID:BDSC_35213 | |
| Genetic reagent (*D. melanogaster*) | UAS-Notch RNAi: y[1] sc[*] v[1] sev[21]; P{y[ + t7.7] v[ + t1.8] = TRiP.GLV21004}attP2 | Bloomington *Drosophila* Stock Center | RRID:BDSC_35620 | |
| Genetic reagent (*D. melanogaster*) | UAS-G-TRACE: w[*]; P{w[ + mC] = UAS-RedStinger}4, P{w[ + mC] = UAS FLP.D}JD1, P{w[ + mC] = Ubi-p63E(FRT.STOP)Stinger}9F6/CyO | Bloomington *Drosophila* Stock Center | RRID:BDSC_28280 | |
| Genetic reagent (*D. melanogaster*) | UAS-G-TRACE: w[*]; P{w[ + mC] = UAS-RedStinger}6, P{w[ + mC] = UAS FLP.Exel}3, P{w[ + mC] = Ubi-p63E(FRT.STOP)Stinger}15F2 | Bloomington *Drosophila* Stock Center | RRID:BDSC_28281 | |
| Genetic reagent (*D. melanogaster*) | UAS-secreted spitz: w[*]; P{w[ + mC] = UAS-sSpiCS}T28 | Bloomington *Drosophila* Stock Center | RRID:BDSC_63134 | |
| Genetic reagent (*D. melanogaster*) | UAS-secreted spitz: w[*]; P{w[ + mC] = UASp spi.sec}3/TM3, Ser[1]() | Bloomington *Drosophila* Stock Center | RRID:BDSC_58436 | |
| Genetic reagent (*D. melanogaster*) | UAS-secreted gurken: w[*]; P{w[ + mC] = UASp grk.sec}2/CyO | Bloomington *Drosophila* Stock Center | RRID:BDSC_58417 | |
| Genetic reagent (*D. melanogaster*) | UAS-gurken ΔTC | PMID: 10559478 | | Dr. Trudi Schüpbach (Princeton University) |
| Genetic reagent (*D. melanogaster*) | UAS- secreted Keren | PMID: 12169630 | | Dr. Matthew Freeman (MRC Laboratory of Molecular Biology) |

*Appendix 1 Continued on next page*

*Appendix 1 Continued*

| Reagent type (species) or resource | Designation | Source or reference | Identifiers | Additional information |
|---|---|---|---|---|
| Genetic reagent (*D. melanogaster*) | UAS-Vein | PMID: 8824589 | | Dr. Amanda Simcox (The Ohio State University) |
| Genetic reagent (*D. melanogaster*) | UAS-Grim | PMID: 9846179 | | Dr. John Nambu (University of Massachusetts) |
| Genetic reagent (*D. melanogaster*) | Egfr(-): Egfr[f24]/T(2;3)TSTL, CyO: TM6B, Tb[1] | Bloomington *Drosophila* Stock Center | RRID:BDSC_6500 | |
| Antibody | (Mouse monoclonal) anti–Fasciclin III (*Drosophila*) | DSHB | Cat#: 7G10; RRID: AB_528238 | IHC (1:50) |
| Antibody | (Mouse monoclonal) anti-phospho-Histone H3 (Ser10) (6G3) | Cell Signaling Technology | Cat#: 9,706 S; RRID:AB_331748 | IHC (1:400) |
| Antibody | (Guinea Pig polyclonal) anti-Traffic Jam | Laboratory of D. Godt (*Li et al., 2003*) | N/A | IHC (1:20,000) |
| Antibody | (Rabbit polyclonal) anti-Vasa (d-260) | Santa Cruz Biotechnology | Cat#: SC-30210; RRID:AB_793874 | IHC (1:200) |
| Antibody | (Chicken polyclonal) anti-GFP | Abcam | Cat#: ab13970; RRID:AB_300798 | IHC (1:10,000) |
| Antibody | (Rabbit polyclonal) anti-DsRed | Takara Bio | Cat#: 632496; RRID:AB_10013483 | IHC (1:10,000) |
| Antibody | (Rabbit polyclonal) anti-dpErk | Cell Signaling Technology | Cat#: 4370; RRID:AB_2315112 | IHC (1:100) |
| Antibody | (Goat polyclonal) anti-Mouse IgG (H + L) secondary antibody, Alexa Fluor 488 conjugate | ThermoFisher Scientific | Cat#: A11029; RRID:AB_2534088 | IHC (1:400) |
| Antibody | (Goat polyclonal) anti-Rabbit IgG (H + L) secondary antibody, Alexa Fluor 568 conjugate | ThermoFisher Scientific | Cat#: A11011; RRID:AB_143157 | IHC (1:200) |
| Antibody | (Goat polyclonal) anti-Chicken IgY (H + L) secondary antibody, Alexa Fluor 488 conjugate | ThermoFisher Scientific | Cat#: A11039; RRID:AB_2534096 | IHC (1:400) |
| Antibody | (Goat polyclonal) anti-Guinea Pig IgG (H + L) secondary antibody, Alexa Fluor 568 conjugate | ThermoFisher Scientific | Cat#: A11075; RRID:AB_2534119 | IHC (1:200) |
| Antibody | (Goat polyclonal) anti-Guinea Pig IgG (H + L) secondary antibody, Alexa Fluor 633 conjugate | ThermoFisher Scientific | Cat#: A21105; RRID:AB_2535757 | IHC (1:200) |
| Chemical compound, drug | 4,6-diamidino-2-phenylindole (DAPI) | Millipore/Sigma (formerly Sigma-Aldrich) | Cat#: 10236276001; CAS: 28718-90-3 | IHC (1 µg/mL) |
| Chemical compound, drug | 16% Paraformaldehyde (formaldehyde) aqueous solution | Electron Microscopy Sciences | Cat#: 15710; CAS: 50-00-0 | |
| Chemical compound, drug | Goat Serum | Millipore/Sigma (formerly Sigma-Aldrich) | Cat#: G9023 | |
| Chemical compound, drug | Vectashield antifade mounting medium | Vector Laboratories | Cat#: H-1000 | |
| Chemical compound, drug | Phosphatase Inhibitor Cocktail 2 | Millipore/Sigma (formerly Sigma-Aldrich) | Cat#: P5726 | |
| Chemical compound, drug | DMSO (Dimethyl Sulfoxide), Sterile | Cell Signaling Technology | Cat#: 12,611 P | |
| Software, algorithm | Fiji | *Schindelin et al., 2012* | https://www.fiji.sc/ | |
| Software, algorithm | Zeiss LSM | Carl Zeiss Microscopy | https://www.zeiss.com/microscopy/us/downloads/lsm-5-series.html | |
| Software, algorithm | Zen | Carl Zeiss Microscopy | https://www.zeiss.com/microscopy/int/products/microscope-software/zen.html | |
| Software, algorithm | Prism 6 | GraphPad | http://www.graphpad.com/scientific-software/prism/ | |

**Appendix 1—table 1.** Screen Summary.

| Reagent type (species) or resource | Designation | Source or reference | Identifiers | Pathway | Additional Information |
|---|---|---|---|---|---|
| Genetic reagent (*D. melanogaster*) | UAS-EcR-RNAi: $y^1$ $v^1$; P{y[ + t7.7] v[ + t1.8] = TRiP.HMC03114}attP2/TM3, Sb$^1$ | Bloomington *Drosophila* Stock Center | RRID:BDSC_50712 | Ecdysone | |
| Genetic reagent (*D. melanogaster*) | UAS-Ecr RNAi: $y^1$ $v^1$; P{y[ + t7.7] v[ + t1.8] = TRiP.HMJ22371}attP40 | Bloomington *Drosophila* Stock Center | RRID:BDSC_58286 | Ecdysone | |
| Genetic reagent (*D. melanogaster*) | UAS-btl-RNAi: $y^1$ sc[*] $v^1$ sev$^{21}$; P{y[ + t7.7] v[ + t1.8] = TRiP.HMS02038}attP2 | Bloomington *Drosophila* Stock Center | RRID:BDSC_40871 | FGFR | |
| Genetic reagent (*D. melanogaster*) | UAS-btl-RNAi: $y^1$ sc[*] $v^1$ sev$^{21}$; P{y[ + t7.7] v[ + t1.8] = TRiP.HMS02656}attP40 | Bloomington *Drosophila* Stock Center | RRID: BDSC_43544 | FGFR | |
| Genetic reagent (*D. melanogaster*) | UAS-btl-RNAi: $y^1$ sc[*] $v^1$ sev$^{21}$; P{y[ + t7.7] v[ + t1.8] = TRiP.HMC04140}attP2 | Bloomington *Drosophila* Stock Center | RRID:BDSC_55870 | FGFR | |
| Genetic reagent (*D. melanogaster*) | UAS-btl-RNAi: $y^1$ $v^1$; P{y[ + t7.7] v[ + t1.8] = TRiP.HMS05005}attP40 | Bloomington *Drosophila* Stock Center | RRID:BDSC_ 60013 | FGFR | |
| Genetic reagent (*D. melanogaster*) | UAS-htl-RNAi: $y^1$ sc[*] $v^1$ sev$^{21}$; P{y[ + t7.7] v[ + t1.8] = TRiP.HMS01437}attP2 | Bloomington *Drosophila* Stock Center | RRID:BDSC_35024 | FGFR | |
| Genetic reagent (*D. melanogaster*) | UAS-htl-RNAi: $y^1$ sc[*] $v^1$ sev$^{21}$; P{y[ + t7.7] v[ + t1.8] = TRiP.HMS04514}attP40 | Bloomington *Drosophila* Stock Center | RRID:BDSC_57313 | FGFR | |
| Genetic reagent (*D. melanogaster*) | UAS-htl-RNAi: $y^1$ $v^1$; P{y[ + t7.7] v[ + t1.8] = TRiP.HMJ22375}attP40 | Bloomington *Drosophila* Stock Center | RRID:BDSC_ 58289 | FGFR | |
| Genetic reagent (*D. melanogaster*) | UAS-Galphaf-RNAi: $y^1$ sc[*] $v^1$ sev$^{21}$; P{y[ + t7.7] v[ + t1.8] = TRiP .GL01545}attP40 | Bloomington *Drosophila* Stock Center | RRID:BDSC_43201 | GPCR | |
| Genetic reagent (*D. melanogaster*) | UAS-Galphai-RNAi: $y^1$ sc[*] $v^1$ sev$^{21}$; P{y[ + t7.7] v[ + t1.8] = TRiP.HMS01273}attP2/ TM3, Sb$^1$ | Bloomington *Drosophila* Stock Center | RRID:BDSC_34924 | GPCR | |
| Genetic reagent (*D. melanogaster*) | UAS-Galphai-RNAi: $y^1$ sc[*] $v^1$ sev$^{21}$; P{y[ + t7.7] v[ + t1.8] = TRiP .GL00328}attP2 | Bloomington *Drosophila* Stock Center | RRID:BDSC_ 35407 | GPCR | |
| Genetic reagent (*D. melanogaster*) | UAS-Galphai-RNAi: $y^1$ $v^1$; P{y[ + t7.7] v[ + t1.8] = TRiP.HMS02138}attP40 | Bloomington *Drosophila* Stock Center | RRID:BDSC_40890 | GPCR | |
| Genetic reagent (*D. melanogaster*) | UAS-Galphao-RNAi: $y^1$ sc[*] $v^1$ sev$^{21}$; P{y[ + t7.7] v[ + t1.8] = TRiP.HMS01129}attP2 | Bloomington *Drosophila* Stock Center | RRID:BDSC_34653 | GPCR | |
| Genetic reagent (*D. melanogaster*) | UAS-Galphaq-RNAi: $y^1$ $v^1$; P{y[ + t7.7] v[ + t1.8] = TRiP.JF02464}attP2 | Bloomington *Drosophila* Stock Center | RRID:BDSC_33765 | GPCR | |
| Genetic reagent (*D. melanogaster*) | UAS-Galphaq-RNAi: $y^1$ sc[*] $v^1$ sev$^{21}$; P{y[ + t7.7] v[ + t1.8] = TRiP.HMS03015}attP2 | Bloomington *Drosophila* Stock Center | RRID:BDSC_36775 | GPCR | |
| Genetic reagent (*D. melanogaster*) | UAS-Galphaq-RNAi: $y^1$ sc[*] $v^1$ sev$^{21}$; P{y[ + t7.7] v[ + t1.8] = TRiP .GL01048}attP2 | Bloomington *Drosophila* Stock Center | RRID:BDSC_36820 | GPCR | |
| Genetic reagent (*D. melanogaster*) | UAS-Galphas-RNAi: $y^1$ $v^1$; P{y[ + t7.7] v[ + t1.8] = TRiP.HMC03106}attP2 | Bloomington *Drosophila* Stock Center | RRID:BDSC_50704 | GPCR | |
| Genetic reagent (*D. melanogaster*) | UAS-ci: w[*]; P{w[ + mC] = UAS ci.HA.wt}3 | Bloomington *Drosophila* Stock Center | RRID:BDSC_32570 | Hedgehog | |
| Genetic reagent (*D. melanogaster*) | UAS-Ci-activated: w[*]; P{w[ + mC] = UAS HA.ci.m1-3*103}2 | Bloomington *Drosophila* Stock Center | RRID:BDSC_32571 | Hedgehog | |
| Genetic reagent (*D. melanogaster*) | UAS-ci-RNAi: $y^1$ sc[*] $v^1$ sev$^{21}$; P{y[ + t7.7] v[ + t1.8] = TRiP.HMC05801}attP2 | Bloomington *Drosophila* Stock Center | RRID:BDSC_64928 | Hedgehog | |
| Genetic reagent (*D. melanogaster*) | UAS-Hmgcr-RNAi: $y^1$ $v^1$; P{y[ + t7.7] v[ + t1.8] = TRiP.HMC03053}attP40 | Bloomington *Drosophila* Stock Center | RRID:BDSC_50652 | Hedgehog | |
| Genetic reagent (*D. melanogaster*) | UAS-hpo-RNAi: $y^1$ $v^1$; P{y[ + t7.7] v[ + t1.8] = TRiP.HMS00006}attP2 | Bloomington *Drosophila* Stock Center | RRID:BDSC_33614 | Hippo | |
| Genetic reagent (*D. melanogaster*) | UAS-hpo-RNAi: $y^1$ sc[*] $v^1$ sev$^{21}$; P{y[ + t7.7] v[ + t1.8] = TRiP .GL00046}attP2 | Bloomington *Drosophila* Stock Center | RRID:BDSC_35176 | Hippo | |
| Genetic reagent (*D. melanogaster*) | UAS-yki-activated: w[*]; P{y[ + t7.7] w[ + mC] = UAS yki.S111A.S168A.S250A.V5} attP2 | Bloomington *Drosophila* Stock Center | RRID:BDSC_28817 | Hippo | |
| Genetic reagent (*D. melanogaster*) | UAS-yki-RNAi: $y^1$ $v^1$; P{y[ + t7.7] v[ + t1.8] = TRiP.HMS00041}attP2 | Bloomington *Drosophila* Stock Center | RRID:BDSC_34067 | Hippo | |
| Genetic reagent (*D. melanogaster*) | UAS-Akt: $y^1$ w[1118]; P{w[ + mC] = UAS Akt.Exel}2 | Bloomington *Drosophila* Stock Center | RRID:BDSC_8191 | InR | |
| Genetic reagent (*D. melanogaster*) | UAS-Akt-activated: w[*]; P{w[ + mC] = UAS-myr-Akt1.V}3/TM3, Sb$^1$ | Bloomington *Drosophila* Stock Center | RRID:BDSC_50758 | InR | No longer available |

*Appendix 1—table 1 Continued on next page*

*Appendix 1—table 1 Continued*

| Reagent type (species) or resource | Designation | Source or reference | Identifiers | Pathway | Additional Information |
|---|---|---|---|---|---|
| Genetic reagent (*D. melanogaster*) | UAS-Akt-RNAi: $y^1$ $v^1$; P{y[ + t7.7] v[ + t1.8] = TRiP.HMS00007}attP2 | Bloomington *Drosophila* Stock Center | RRID:BDSC_33615 | InR | |
| Genetic reagent (*D. melanogaster*) | UAS-Akt-RNAi: $y^1$ $v^1$; P{y[ + t7.7] v[ + t1.8] = TRiP.JF02668}attP2 | Bloomington *Drosophila* Stock Center | RRID:BDSC_27518 | InR | |
| Genetic reagent (*D. melanogaster*) | UAS-Foxo: $y^1$ w[*]$^†$; P{w[ + mC] = UAS foxo.P}2 | Bloomington *Drosophila* Stock Center | RRID:BDSC_9575 | InR | |
| Genetic reagent (*D. melanogaster*) | UAS-Foxo: w[1118]; P{w[ + mC] = UASp foxo.S}3 | Bloomington *Drosophila* Stock Center | RRID:BDSC_42221 | InR | |
| Genetic reagent (*D. melanogaster*) | UAS-Foxo: w[1118]; P{w[ + mC] = UASp foxo.GFP}3 | Bloomington *Drosophila* Stock Center | RRID:BDSC_43633 | InR | |
| Genetic reagent (*D. melanogaster*) | UAS-Foxo: w[1118]; P{w[ + mC] = UASp foxo.GFP}2 | Bloomington *Drosophila* Stock Center | RRID:BDSC_44214 | InR | |
| Genetic reagent (*D. melanogaster*) | UAS-Foxo RNAi: y[1] sc[*] v[1] sev[21]; P{y[ + t7.7] v[ + t1.8] = TRiP.HMS00422}attP2 | Bloomington *Drosophila* Stock Center | RRID:BDSC_32427 | InR | |
| Genetic reagent (*D. melanogaster*) | UAS-Foxo RNAi: $y^1$ sc[*] $v^1$ sev$^{21}$; P{y[ + t7.7] v[ + t1.8] = TRiP.HMS00793}attP2 | Bloomington *Drosophila* Stock Center | RRID:BDSC_32993 | InR | |
| Genetic reagent (*D. melanogaster*) | UAS-InR-DN: $y^1$ w[1118]; P{w[ + mC] = UAS InR.K1409A}2 | Bloomington *Drosophila* Stock Center | RRID:BDSC_8252 | InR | |
| Genetic reagent (*D. melanogaster*) | UAS-InR-DN: $y^1$ w[1118]; P{w[ + mC] = UAS InR.K1409A}3 | Bloomington *Drosophila* Stock Center | RRID:BDSC_8253 | InR | |
| Genetic reagent (*D. melanogaster*) | UAS-InR-RNAi: $y^1$ $v^1$; P{y[ + t7.7] v[ + t1.8] = TRiP.HMS03166}attP40 | Bloomington *Drosophila* Stock Center | RRID:BDSC_51518 | InR | |
| Genetic reagent (*D. melanogaster*) | UAS-Pi3K21B: $y^1$ w[*]; P{w[ + mC] = UAS-Pi3K21B.HA}2 | Bloomington *Drosophila* Stock Center | RRID:BDSC_25899 | InR | |
| Genetic reagent (*D. melanogaster*) | UAS-dome-RNAi: w[1118]; P{GD14494} v36356 | Vienna *Drosophila* Resource Center | Stock #: 36356 | Jak-Stat | |
| Genetic reagent (*D. melanogaster*) | UAS-dome-RNAi: P{KK104700}VIE-260B | Vienna *Drosophila* Resource Center | Stock #: 106071 | Jak-Stat | |
| Genetic reagent (*D. melanogaster*) | UAS-hop-activated: w; UAS-hop[TumL]/CyO | PMID: 7796812 | | Jak-Stat | Dr. Norbert Perrimon (Harvard Medical School) |
| Genetic reagent (*D. melanogaster*) | UAS-Stat92E-RNAi: w[1118]; P{GD4492} v43866 | Vienna *Drosophila* Resource Center | Stock #: 43866 | Jak-Stat | |
| Genetic reagent (*D. melanogaster*) | UAS-Stat92E-RNAi: P{KK100519}VIE-260B | Vienna *Drosophila* Resource Center | Stock #: 106980 | Jak-Stat | |
| Genetic reagent (*D. melanogaster*) | UAS-upd1 | PMID: 10346822 | | Jak-Stat | Dr. David Strutt (Harvard Medical School) |
| Genetic reagent (*D. melanogaster*) | UAS-upd1-RNAi: w[1118]; P{GD1158} v3282 | | Stock #: 3282 | Jak-Stat | |
| Genetic reagent (*D. melanogaster*) | UAS-upd1-RNAi: y[1] sc[*] v[1] sev[21]; P{y[ + t7.7] v[ + t1.8] = TRiP.HMS00545}attP2 | Bloomington *Drosophila* Stock Center | RRID:BDSC_33680 | Jak-Stat | |
| Genetic reagent (*D. melanogaster*) | UAS-upd2-RNAi: $y^1$ sc[*] $v^1$ sev$^{21}$; P{y[ + t7.7] v[ + t1.8] = TRiP.HMS00901}attP2 | Bloomington *Drosophila* Stock Center | RRID:BDSC_33949 | Jak-Stat | |
| Genetic reagent (*D. melanogaster*) | UAS-upd2-RNAi:$^1$ sc[*] $v^1$ sev$^{21}$; P{y[ + t7.7] v[ + t1.8] = TRiP.HMS00948}attP2 | Bloomington *Drosophila* Stock Center | RRID:BDSC_33988 | Jak-Stat | |
| Genetic reagent (*D. melanogaster*) | UAS-upd3-RNAi: $y^1$ sc[ *]$^‡$ $v^1$ sev$^{21}$; P{y[ + t7.7] v[ + t1.8] = TRiP.HMS00646}attP2 | Bloomington *Drosophila* Stock Center | RRID:BDSC_32859 | Jak-Stat | |
| Genetic reagent (*D. melanogaster*) | UAS-hep-activated: $y^1$ w[1118]; P{w[ + mC] = UAS hep.CA}4 | Bloomington *Drosophila* Stock Center | RRID:BDSC_6406 | Jnk | |
| Genetic reagent (*D. melanogaster*) | UAS-hep-activated: w[*]; P{w[ + mC] = UAS Hep.Act}2 | Bloomington *Drosophila* Stock Center | RRID:BDSC_9306 | Jnk | |
| Genetic reagent (*D. melanogaster*) | UAS-kay: w[1118]; P{w[ + mC] = UAS-Fra}2 | Bloomington *Drosophila* Stock Center | RRID:BDSC_7213 | Jnk | |

*Appendix 1—table 1 Continued on next page*

*Appendix 1—table 1 Continued*

| Reagent type (species) or resource | Designation | Source or reference | Identifiers | Pathway | Additional Information |
|---|---|---|---|---|---|
| Genetic reagent (D. melanogaster) | UAS-kay-DN: w[1118]; P{w[ + mC] = UAS Fra.Fbz}5 | Bloomington Drosophila Stock Center | RRID:BDSC_7214 | Jnk | |
| Genetic reagent (D. melanogaster) | UAS-kay-DN: y[1] w[1118]; P{w[ + mC] = UAS Fra.Fbz}7 | Bloomington Drosophila Stock Center | RRID:BDSC_7215 | Jnk | |
| Genetic reagent (D. melanogaster) | UAS-jra: y[1] w[1118]; P{w[ + mC] = UAS-Jra}2 | Bloomington Drosophila Stock Center | RRID:BDSC_7216 | Jnk | |
| Genetic reagent (D. melanogaster) | UAS-kay-RNAi: y[1] sc[*] v[1] sev[21]; P{y[ + t7.7] v[ + t1.8] = TRiP.HMS00254}attP2 | Bloomington Drosophila Stock Center | RRID:BDSC_33379 | Jnk | |
| Genetic reagent (D. melanogaster) | UAS-wgn-RNAi: y[1] sc[*] v[1] sev[21]; P{y[ + t7.7] v[ + t1.8] = TRiP.HMC03962}attP40 | Bloomington Drosophila Stock Center | RRID:BDSC_55275 | Jnk | |
| Genetic reagent (D. melanogaster) | UAS-egr-RNAi: y[1] sc[*] v[1] sev[21]; P{y[ + t7.7] v[ + t1.8] = TRiP.HMC03963}attP40 | Bloomington Drosophila Stock Center | RRID:BDSC_55276 | Jnk | |
| Genetic reagent (D. melanogaster) | UAS-Pvr-RNAi: y[1] sc[*] v[1] sev[21]; P{y[ + t7.7] v[ + t1.8] = TRiP.HMS01662}attP40 | Bloomington Drosophila Stock Center | RRID:BDSC_37520 | Pvr | |
| Genetic reagent (D. melanogaster) | UAS-Pvf1-RNAi: y[1] sc[*] v[1] sev[21]; P{y[ + t7.7] v[ + t1.8] = TRiP.HMS01958}attP40 | Bloomington Drosophila Stock Center | RRID:BDSC_39038 | Pvr | |
| Genetic reagent (D. melanogaster) | UAS-Pvr-DN: w[1118]; P{w[ + mC] = UASp Pvr.DN}D1/CyO | Bloomington Drosophila Stock Center | RRID:BDSC_58430 | Pvr | |
| Genetic reagent (D. melanogaster) | UAS-Pvr-DN: w[1118]; P{w[ + mC] = UASp Pvr.DN}D7 | Bloomington Drosophila Stock Center | RRID:BDSC_58431 | Pvr | |
| Genetic reagent (D. melanogaster) | UAS-aop: w[*]; P{w[ + mC] = UAS aop.WT}Ia/CyO | Bloomington Drosophila Stock Center | RRID:BDSC_5790 | RTK | |
| Genetic reagent (D. melanogaster) | UAS-aop-activated: w[*]; P{w[ + mC] = UAS aop.ACT}IIa | Bloomington Drosophila Stock Center | RRID:BDSC_5789 | RTK | |
| Genetic reagent (D. melanogaster) | UAS-pnt-RNAi: y[1] sc[*] v[1] sev[21]; P{y[ + t7.7] v[ + t1.8] = TRiP.HMS01452}attP2 | Bloomington Drosophila Stock Center | RRID:BDSC_35038 | RTK | |
| Genetic reagent (D. melanogaster) | UAS-rl-RNAi: y[1] sc[*] v[1] sev[21]; P{y[ + t7.7] v[ + t1.8] = TRiP.HMS00173}attP2 | Bloomington Drosophila Stock Center | RRID:BDSC_34855 | RTK | |
| Genetic reagent (D. melanogaster) | UAS-rl-RNAi: y[1] sc[*] v[1] sev[21]; P{y[ + t7.7] v[ + t1.8] = TRiP .GL00215}attP2 | Bloomington Drosophila Stock Center | RRID:BDSC_36058 | RTK | |
| Genetic reagent (D. melanogaster) | UAS-sev-RNAi: y[1] v[1]; P{y[ + t7.7] v[ + t1.8] = TRiP.JF02393}attP2 | Bloomington Drosophila Stock Center | RRID:BDSC_36778 | RTK | |
| Genetic reagent (D. melanogaster) | UAS-sev-RNAi: y[1] v[1]; P{y[ + t7.7] v[ + t1.8] = TRiP.HMC04136}attP2 | Bloomington Drosophila Stock Center | RRID:BDSC_55866 | RTK | |
| Genetic reagent (D. melanogaster) | UAS-tor-RNAi: y[1] v[1]; P{y[ + t7.7] v[ + t1.8] = TRiP.HMS00021}attP2 | Bloomington Drosophila Stock Center | RRID:BDSC_33627 | RTK | |
| Genetic reagent (D. melanogaster) | UAS-tor-RNAi: y[1] v[1]; P{y[ + t7.7] v[ + t1.8] = TRiP.HMJ22419}attP40 | Bloomington Drosophila Stock Center | RRID:BDSC_58312 | RTK | |
| Genetic reagent (D. melanogaster) | UAS-Mad: P{ry[ + t7.2] = hsFLP}12, y[1] w[*]; P{w[ + mC] = UAS Mad.FLAG}2; P{y[ + t7.7] w[ + mC] = mir-ban-lacZ.brC12}attP2 | Bloomington Drosophila Stock Center | RRID:BDSC_44256 | TGFβ | |
| Genetic reagent (D. melanogaster) | UAS-Mad-RNAi: y[1] sc[*] v[1] sev[21]; P{y[ + t7.7] v[ + t1.8] = TRiP.GLV21013}attP2/TM3, Sb[1] | Bloomington Drosophila Stock Center | RRID:BDSC_35648 | TGFβ | |
| Genetic reagent (D. melanogaster) | UAS-Mad-RNAi: y[1] sc[*] v[1] sev[21]; P{y[ + t7.7] v[ + t1.8] = TRiP .GL01527}attP40 | Bloomington Drosophila Stock Center | RRID:BDSC_43183 | TGFβ | |
| Genetic reagent (D. melanogaster) | UAS-Smox-RNAi: [1]sc[*] v[1] sev[21]; P{y[ + t7.7] v[ + t1.8] = TRiP.HMS02203}attP40 | Bloomington Drosophila Stock Center | RRID:BDSC_41670 | TGFβ | |
| Genetic reagent (D. melanogaster) | UAS-Smox-RNAi: y[1] v[1]; P{y[ + t7.7] v[ + t1.8] = TRiP .GL01476}attP2 | Bloomington Drosophila Stock Center | RRID:BDSC_43138 | TGFβ | |
| Genetic reagent (D. melanogaster) | UAS-cact-RNAi: y[1] sc[*] v[1] sev[21]; P{y[ + t7.7] v[ + t1.8] = TRiP.HMS00084}attP2 | Bloomington Drosophila Stock Center | RRID:BDSC_34775 | Toll | |
| Genetic reagent (D. melanogaster) | UAS-cact-RNAi: y[1] sc[*] v[1] sev[21]; P{y[ + t7.7] v[ + t1.8] = TRiP .GL00627}attP40 | Bloomington Drosophila Stock Center | RRID:BDSC_37484 | Toll | |

*Appendix 1—table 1 Continued on next page*

*Appendix 1—table 1 Continued*

| Reagent type (species) or resource | Designation | Source or reference | Identifiers | Pathway | Additional Information |
|---|---|---|---|---|---|
| Genetic reagent (*D. melanogaster*) | UAS-dl: y[1] w[*]; P{w[ + mC] = UAS dl.H}2 | Bloomington *Drosophila* Stock Center | RRID:BDSC_9319 | Toll | |
| Genetic reagent (*D. melanogaster*) | UAS-dl-RNAi: y[1] sc[*] v[1] sev[21]; P{y[ + t7.7] v[ + t1.8] = TRiP.HMS00727}attP2 | Bloomington *Drosophila* Stock Center | RRID:BDSC_32934 | Toll | |
| Genetic reagent (*D. melanogaster*) | UAS-dl-RNAi: y[1]sc[*] v[1] sev[21]; P{y[ + t7.7] v[ + t1.8] = TRiP.HMS00028}attP2 | Bloomington *Drosophila* Stock Center | RRID:BDSC_34938 | Toll | |
| Genetic reagent (*D. melanogaster*) | UAS-dl-RNAi: y[1] sc[*] v[1] sev[21]; P{y[ + t7.7] v[ + t1.8] = TRiP .GL00610}attP40 | Bloomington *Drosophila* Stock Center | RRID:BDSC_36650 | Toll | |
| Genetic reagent (*D. melanogaster*) | UAS-dl-RNAi: y[1] sc[*] v[1] sev[21]; P{y[ + t7.7] v[ + t1.8] = TRiP .GL00676}attP2 | Bloomington *Drosophila* Stock Center | RRID:BDSC_38905 | Toll | |
| Genetic reagent (*D. melanogaster*) | UAS-arm-RNAi: y[1] sc[*] v[1] sev[21]; P{y[ + t7.7] v[ + t1.8] = TRiP.HMS01414}attP2 | Bloomington *Drosophila* Stock Center | RRID:BDSC_35004 | Wnt | |
| Genetic reagent (*D. melanogaster*) | UAS-pan: y[1] w[1118]; P{w[ + mC] = UAS pan.dTCF}24/CyO | Bloomington *Drosophila* Stock Center | RRID:BDSC_4837 | Wnt | |
| Genetic reagent (*D. melanogaster*) | UAS-pan: y[1] w[1118]; P{w[ + mC] = UAS pan.dTCF}4 | Bloomington *Drosophila* Stock Center | RRID:BDSC_4838 | Wnt | |
| Genetic reagent (*D. melanogaster*) | UAS-pan-constitutive repressor: y[1] w[1118]; P{w[ + mC] = UAS pan.dTCFDeltaN}4 | Bloomington *Drosophila* Stock Center | RRID:BDSC_4784 | Wnt | |
| Genetic reagent (*D. melanogaster*) | UAS-pan-constitutive repressor: y[1] w[1118]; P{w[ + mC] = UAS pan.dTCFDeltaN}5 | Bloomington *Drosophila* Stock Center | RRID:BDSC_4785 | Wnt | |
| Genetic reagent (*D. melanogaster*) | UAS-pan-RNAi: y v; P{y[ + t7.7] v[ + t1.8] = TRiP.HMS02015}attP40/CyO | Bloomington *Drosophila* Stock Center | RRID:BDSC_40848 | Wnt | |
| Genetic reagent (*D. melanogaster*) | UAS-wg-RNAi: w[1118]; P{GD5007}v13352 | Vienna *Drosophila* Resource Center | Stock #: 13352 | Wnt | |

*n = at least 20–199 testes for all lines.

†lines listed in the key resource table are not repeated here.

‡no lines listed here had significant numbers of dividing hub cells compared to UAS-GFP-RNAi controls.

