## [Editor Report]

In this manuscript, the Authors demonstrate that EGFR signaling plays an important role in somatic cyst stem cells and hub cells in the male germ cell lineage. Using a variety of genetic, biochemical and cell biological approaches, they provide a regulatory frame work for how the hub cells maintain their cell fate.

---

## [Decision Letter]

**Decision letter after peer review:**

Thank you for sending your article entitled "Activation of the EGFR/MAPK pathway drives transdifferentiation of quiescent niche cells to stem cells in the *Drosophila* testis" for peer review at *eLife*. Your article is being evaluated by 3 peer reviewers, and the evaluation is being overseen by a Reviewing Editor and Utpal Banerjee as the Senior Editor.

Summary:

Several strengths of the manuscript were noted by the Reviewers including well designed experiments, use of the appropriate fly genetics tools, well described transdifferentiation phenotype of quiescent niche cells into stem cells in the fly testis and elucidating the role of EGFR signaling in this process. The Reviewers also suggested critical experiments that were missing. Some of these include the explanation for relatively low % of hub cells activated by EGFR signaling that enter into mitosis, and knocking down some of the other negative regulators (such as PTEN and others) that could compensate for over expression of EGFR pathway. Rev 3 also raised a concern that there is not sufficient evidence to localize the source of EGFs to the germ cell population.

Based on these and their other more detailed comments (described below), I am recommending a revision of your manuscript.

*Reviewer #1:*

They first showed that, during steady state, EGRF signaling is suppressed in the hub cells, maintaining these cells quiescent, while its forced activation (e.g., by means of induction of activated form of the receptor and inhibiting intracellular suppressors) results in the cell cycle activation and transdifferentiation to CySCs. They also showed that EGFR signaling plays vital roles in the previously reported transdifferentiation process triggered by the ablation of CySCs and/or early cyst cells, by means of changing signal strength using EGFR heterozygotes.

The authors further demonstrated that suppression of a secreted inhibitor of EGF ligands (i.e., Argos) elicits the hub-to-CySC transdifferentiation. Based on this finding, in combination with the previous observations that EGF ligands secreted from the germ cells (presumptively GSCs) induces the encapsulation of germ cells by cyst cells, and that germ cells are indispensable for the recovery of depleted CySCs (most likely through transdifferentiation), they have proposed an interesting model. In particular, they hypothesize that, in steady state, although GSC-derived EGFR ligands act on the cyst (stem) cells to promote encapsulation, their action does not reach the hub cells through the function of Argos (secreted by CySCs), suppressing the cell cycle entry and transdifferentiation program in the hub. However, when CySCs are ablated, the loss of Argos results in the activation of proliferation and transdifferentiation program in the hub cells to replenish the lost CySCs.

Throughout, this study has been conducted thoroughly and well documented, and the proposed model is interesting. Whilst, although series of genetical analyses consistently support this hypothesis, it remains puzzling what underpins the differential interplay between EGFR ligands and Argos between different target cells. In particular, while Argos inhibits the effect of the ligands on the hub cells, it appears not to affect the action on CySCs. I am fully convinced that this model is supported from a genetics point of view, but there should be some additional mechanism that remains to be elucidated. It would make this study really a strong piece of work, beyond an increment of their previous study, if some insights has been made to this remaining question. Naively, I would suspect that some spatial (local) separation of the ligands, EGFR, and the inhibitor (Argos) could explain this puzzle. For example, on the CySCs, EGFR could be displayed on their surface toward GSCs, while Argos could be secreted to the opposite, outside the capsule to antagonize the leaked ligands, and/or to the site of GSC-hub interaction. The localization of EGFR on the hub could be an important component of such differential action. Is it ever possible to investigate the local distribution of these factors in the undisturbed and CySC-depleted gonads?

Another point is that the conclusion that EGFR signaling is NOT involved in the formation of ectopic hub following the genetic ablation of CySCs, another striking discovery by the authors, is not convincing. The authors only showed that the NULL HYPOTHESIS that the induction efficiency of ectopic hub is not affected by suppressing the EGFR signaling was NOT REJECTED statistically. Of course, this results does not support the afore-mentioned conclusion. Indeed, the actual values of the occurrence of ectopic hubs was lower in the EGFR heterozygous background compared with the wild-type situation (19% vs 32%, albeit the non-significant statistical difference). Based on the current dataset and analyses, the authors may not be able to draw any conclusions with regard to the involvement of EGFR signaling in the ectopic hub formation. Further data collection and/or more careful statistical analyses would be essential to discuss this issue appropriately.

*Reviewer #2:*

The manuscript by Greenspan et al., characterizes the role of Epidermal growth factor receptor (EGFR) signaling in the transdifferentiation of hub cells in the *Drosophila* testis stem cell niche, facilitating the replenishment of somatic cyst stem cells after ablation. The myriad of genetic manipulation approaches that the authors use to demonstrate the involvement of the EGFR pathway in transdifferentiation in vivo is truly impressive, and provides a convincing picture that supports the overall conclusions.

Some points that require clarification / suggestions to strengthen the manuscript:

– It is not clear as to why, if EGFR is the master regulator of this transactivation process, the percentage of testes with hub cells recruited into mitosis is lower following EGFR activation (4%) than with activation of downstream effectors/transcription factors (i.e. Pointed and Ras – up to 11% mitotic activation). Also, it is noted that during recovery from CySC ablation, 6-10% of testes have mitotic hub cells, again a higher percentage than following EGFR activation. This highlights the possibility that there may be other signaling pathways at play that could act on similar effector molecules.

– The authors note that a large-scale genetic screen was conducted to mis-express components of numerous major signaling pathways and subsequently identify resumption of mitosis in quiescent hub cells. The 'negative' results of this screen are eluded to several times throughout the manuscript, however a summary of the data is not shown. A summary of the genes / pathways that were mis-expressed would be a useful resource to include in the paper, even if this manipulation had no effect on hub-cell transdifferentiation.

– Why is significant mitotic activation seen in hub cells after activation of Egfr type II but not type I? Would perhaps be useful to provide some explanation in the text.

– Activation of EGFR is less effective at driving hub cells into mitosis then the activation of Pointed or Ras, but activation of EGFR seems to be more effective at instigating transdifferentiation in G-TRACE experiments. Can the authors speculate on what the cause of this discrepancy may be?

– Over-expression of EGF ligands was shown to be insufficient to trigger loss of hub-cell quiescence and the authors hypothesise that this could be because negative regulators of the EGF pathway (Sprouty, PTEN, Argos) override this effect. Would it be possible to test this hypothesis by knocking down inhibitor(s) while simultaneously over-expression ligand(s) to show that the effect on hub cell transdifferentiation is increased when compared to inhibitor knockdown alone?

*Reviewer #3:*

The reported findings represent exciting new advances to the field in our understanding of the regulation of homeostasis and regeneration. The authors have utilized genetic approaches to systematically dissect EGF signaling among the hub and CySC populations, including highly complex multi-gene approaches. However, there are some missing controls that would support conclusions that MAPK signaling downstream EGFR participates in the exit from quiescence. Furthermore, the authors assume EGF ligands derive from the germ cell population (portrayed in Figure 5), but the source of EGF ligands remains undetermined. Finally, the participation of PTEN in both hub proliferative activation and transdifferentiation may also suggest that additional pathways may participate in both processes, particularly in the absence of evidence that PTEN promotes EGFR degeneration within the hub. Collectively, however, the studies are of high quality and well written in a manner appropriate for the journal.

– The authors need to better connect activation of the MAPK pathway to proliferative and transdifferentiation responses in the hub. A simple control can be staining Argos, PTEN, and Sprouty knockdowns for dpERK to determine whether or not activation of the MAPK pathway coincides with proliferation and transdifferentiation (in the case with PTEN and Sprout knockdowns only).

– The authors indicate the inhibition of EGF signaling at 3 locations, externally via Argos and inside the hub cell via PTEN or Sprouty, suppresses the onset of hub proliferation and transdifferentiation. Each target individually contributes to the inhibition, suggesting that either multiple upstream signals converge on the MAPK pathway or the three components act synergistically to inhibit the MAPK pathway. For example, with PTEN knockdown, proliferation and transdifferentiation are induced with Sprouty intact. Similarly, in the absence of Argos in CySCs, proliferation is still triggered despite intact downstream inhibition via PTEN and Sprouty. To disseminate whether the inhibitors act in conjunction or from separate pathways, the authors need to perform compound knockdowns of paired targets (i.e. Sprouty + PTEN). Understandably for technical reasons, combined knockdowns of Argos and Sprouty or PTEN are not feasible.

– In the absence of proliferation induction with EGF ligand overexpression from either hub cells or CySCs, there is not sufficient evidence to localize the source of EGFs to the germ cell population (depicted in Figure 5), even with circumstantial evidence from regeneration experiments in Figure 4 that illustrate the presence of germ cells while the hub regenerates. To localize the source of EGF ligands to germ cells given limitations noted in the study, the authors could perform compound ablation using both the C587-Gal4 driver alongside a germ-cell specific driver.

---

## [Author Response]

Reviewer #1:They first showed that, during steady state, EGRF signaling is suppressed in the hub cells, maintaining these cells quiescent, while its forced activation (e.g., by means of induction of activated form of the receptor and inhibiting intracellular suppressors) results in the cell cycle activation and transdifferentiation to CySCs. They also showed that EGFR signaling plays vital roles in the previously reported transdifferentiation process triggered by the ablation of CySCs and/or early cyst cells, by means of changing signal strength using EGFR heterozygotes.The authors further demonstrated that suppression of a secreted inhibitor of EGF ligands (i.e., Argos) elicits the hub-to-CySC transdifferentiation. Based on this finding, in combination with the previous observations that EGF ligands secreted from the germ cells (presumptively GSCs) induces the encapsulation of germ cells by cyst cells, and that germ cells are indispensable for the recovery of depleted CySCs (most likely through transdifferentiation), they have proposed an interesting model. In particular, they hypothesize that, in steady state, although GSC-derived EGFR ligands act on the cyst (stem) cells to promote encapsulation, their action does not reach the hub cells through the function of Argos (secreted by CySCs), suppressing the cell cycle entry and transdifferentiation program in the hub. However, when CySCs are ablated, the loss of Argos results in the activation of proliferation and transdifferentiation program in the hub cells to replenish the lost CySCs.Throughout, this study has been conducted thoroughly and well documented, and the proposed model is interesting. Whilst, although series of genetical analyses consistently support this hypothesis, it remains puzzling what underpins the differential interplay between EGFR ligands and Argos between different target cells. In particular, while Argos inhibits the effect of the ligands on the hub cells, it appears not to affect the action on CySCs. I am fully convinced that this model is supported from a genetics point of view, but there should be some additional mechanism that remains to be elucidated. It would make this study really a strong piece of work, beyond an increment of their previous study, if some insights has been made to this remaining question. Naively, I would suspect that some spatial (local) separation of the ligands, EGFR, and the inhibitor (Argos) could explain this puzzle. For example, on the CySCs, EGFR could be displayed on their surface toward GSCs, while Argos could be secreted to the opposite, outside the capsule to antagonize the leaked ligands, and/or to the site of GSC-hub interaction. The localization of EGFR on the hub could be an important component of such differential action. Is it ever possible to investigate the local distribution of these factors in the undisturbed and CySC-depleted gonads?

We tested several transgene reagents, such as EGFR-GFP, which unfortunately did not reveal receptor distribution in our system. We also tried immunostaining for EGFR, Spitz, and Argos, but as is often the case for secreted factors and their cell-surface receptors, we were not able to detect a reliable signal above background. Building additional reagents to test receptor localization is beyond the scope of this current study, but these are interesting questions for future studies.

Another point is that the conclusion that EGFR signaling is NOT involved in the formation of ectopic hub following the genetic ablation of CySCs, another striking discovery by the authors, is not convincing. The authors only showed that the NULL HYPOTHESIS that the induction efficiency of ectopic hub is not affected by suppressing the EGFR signaling was NOT REJECTED statistically. Of course, this results does not support the afore-mentioned conclusion. Indeed, the actual values of the occurrence of ectopic hubs was lower in the EGFR heterozygous background compared with the wild-type situation (19% vs 32%, albeit the non-significant statistical difference). Based on the current dataset and analyses, the authors may not be able to draw any conclusions with regard to the involvement of EGFR signaling in the ectopic hub formation. Further data collection and/or more careful statistical analyses would be essential to discuss this issue appropriately.

We thank the reviewer for encouraging us to repeat this experiment. Upon analyzing many more testes, we found that ectopic niche formation was very significantly reduced in testes lacking one copy of Egfr compared to control testes. We have added these new data to Table 6 and revised the text accordingly.

Reviewer #2:The manuscript by Greenspan et al., characterizes the role of Epidermal growth factor receptor (EGFR) signaling in the transdifferentiation of hub cells in the *Drosophila* testis stem cell niche, facilitating the replenishment of somatic cyst stem cells after ablation. The myriad of genetic manipulation approaches that the authors use to demonstrate the involvement of the EGFR pathway in transdifferentiation in vivo is truly impressive, and provides a convincing picture that supports the overall conclusions.Some points that require clarification / suggestions to strengthen the manuscript:– It is not clear as to why, if EGFR is the master regulator of this transactivation process, the percentage of testes with hub cells recruited into mitosis is lower following EGFR activation (4%) than with activation of downstream effectors/transcription factors (i.e. Pointed and Ras – up to 11% mitotic activation). Also, it is noted that during recovery from CySC ablation, 6-10% of testes have mitotic hub cells, again a higher percentage than following EGFR activation. This highlights the possibility that there may be other signaling pathways at play that could act on similar effector molecules.

The apparent differences in the percentage of positive testes upon activation of EGFR vs. downstream pathway members are not significant, but we agree that other signaling pathways could be involved in recovery of CySCs after ablation. We have clarified this point in the Discussion.

– The authors note that a large-scale genetic screen was conducted to mis-express components of numerous major signaling pathways and subsequently identify resumption of mitosis in quiescent hub cells. The 'negative' results of this screen are eluded to several times throughout the manuscript, however a summary of the data is not shown. A summary of the genes / pathways that were mis-expressed would be a useful resource to include in the paper, even if this manipulation had no effect on hub-cell transdifferentiation.

We thank the reviewer for this suggestion. We have added a summary of the genes and pathways tested in our screen (Appendix 1).

– Why is significant mitotic activation seen in hub cells after activation of Egfr type II but not type I? Would perhaps be useful to provide some explanation in the text.

We have not independently tested the strength of these constructs, so the reason for the difference is not clear.

– Activation of EGFR is less effective at driving hub cells into mitosis then the activation of Pointed or Ras, but activation of EGFR seems to be more effective at instigating transdifferentiation in G-TRACE experiments. Can the authors speculate on what the cause of this discrepancy may be?

As noted above, the differences in percentage of testes with mitotic hub cells (Table 1) are not statistically significant. In the G-TRACE experiments, the percentages of positive testes are significantly different between EGFR, P1, and P2 (p < 0.05 by ANOVA). However, it’s not clear whether the differences are biologically relevant or reflect differences in transgene strength, genetic background (with or without the G-TRACE system), or some other factor, and teasing apart these potential factors would require experiments beyond the scope of the current study.

– Over-expression of EGF ligands was shown to be insufficient to trigger loss of hub-cell quiescence and the authors hypothesise that this could be because negative regulators of the EGF pathway (Sprouty, PTEN, Argos) override this effect. Would it be possible to test this hypothesis by knocking down inhibitor(s) while simultaneously over-expression ligand(s) to show that the effect on hub cell transdifferentiation is increased when compared to inhibitor knockdown alone?

We thank the reviewer for this suggestion. To address this question, we used the conditional cyst lineage driver c587-Gal4; Gal80[ts] to knock down Argos and over-express Spitz concurrently. Contrary to our expectations, we found that the percentage of testes with dividing hub cells in these flies was not significantly different than in flies with Argos knockdown alone. We have added these data to Table 4, but as we discuss in the text, the reason for this result is not clear.

Reviewer #3:The reported findings represent exciting new advances to the field in our understanding of the regulation of homeostasis and regeneration. The authors have utilized genetic approaches to systematically dissect EGF signaling among the hub and CySC populations, including highly complex multi-gene approaches. However, there are some missing controls that would support conclusions that MAPK signaling downstream EGFR participates in the exit from quiescence. Furthermore, the authors assume EGF ligands derive from the germ cell population (portrayed in Figure 5), but the source of EGF ligands remains undetermined. Finally, the participation of PTEN in both hub proliferative activation and transdifferentiation may also suggest that additional pathways may participate in both processes, particularly in the absence of evidence that PTEN promotes EGFR degeneration within the hub. Collectively, however, the studies are of high quality and well written in a manner appropriate for the journal.– The authors need to better connect activation of the MAPK pathway to proliferative and transdifferentiation responses in the hub. A simple control can be staining Argos, PTEN, and Sprouty knockdowns for dpERK to determine whether or not activation of the MAPK pathway coincides with proliferation and transdifferentiation (in the case with PTEN and Sprout knockdowns only).

We thank the reviewer for this suggestion. We assessed dpERK immunostaining levels upon conditional knockdown of PTEN or Sprouty in the hub and compared them to control flies (y w) with no EGFR pathway perturbation. We found that dpERK levels in hub cells were significantly elevated upon knockdown of PTEN or Sprouty, consistent with activation of the MAPK pathway, and we have added these new data to Figure 3. We tried to assess dpERK levels upon knockdown of Argos in cyst lineage cells, but for reasons that are not clear, we got very inconsistent results for this combination and for the cyst lineage driver alone, and we have not included these data.

– The authors indicate the inhibition of EGF signaling at 3 locations, externally via Argos and inside the hub cell via PTEN or Sprouty, suppresses the onset of hub proliferation and transdifferentiation. Each target individually contributes to the inhibition, suggesting that either multiple upstream signals converge on the MAPK pathway or the three components act synergistically to inhibit the MAPK pathway. For example, with PTEN knockdown, proliferation and transdifferentiation are induced with Sprouty intact. Similarly, in the absence of Argos in CySCs, proliferation is still triggered despite intact downstream inhibition via PTEN and Sprouty. To disseminate whether the inhibitors act in conjunction or from separate pathways, the authors need to perform compound knockdowns of paired targets (i.e. Sprouty + PTEN). Understandably for technical reasons, combined knockdowns of Argos and Sprouty or PTEN are not feasible.

While we agree that this is an interesting question, we believe that the results of such an experiment would be difficult to interpret. These knockdowns are dependent on RNAi constructs, which (unlike null alleles) do not result in complete knockdown of a gene. Therefore, if paired targets were to result in a stronger phenotype, we could not distinguish between stronger knockdown of inhibitors in one pathway vs. combined partial knockdown of two pathways.

– In the absence of proliferation induction with EGF ligand overexpression from either hub cells or CySCs, there is not sufficient evidence to localize the source of EGFs to the germ cell population (depicted in Figure 5), even with circumstantial evidence from regeneration experiments in Figure 4 that illustrate the presence of germ cells while the hub regenerates. To localize the source of EGF ligands to germ cells given limitations noted in the study, the authors could perform compound ablation using both the C587-Gal4 driver alongside a germ-cell specific driver.

While we agree that the compound ablation would be a nice experiment to add, we have found that germ cells and cyst cells do not respond equally well to the same death-inducing transgenes, and building flies with all the required transgenes is technically challenging. Fortunately, germ cells are already known to be a source of the EGF ligand Spitz (Sarkar et al., 2007), which we have clarified in the text.